# Calcium transients regulate the apical emergence of basally located progenitors during Xenopus skin development

Neophytos Christodoulou ✉ & Paris A. Skourides ✉

The integration of basally located progenitors into an existing epithelium, termed apical emergence, is crucial for the morphogenesis and homeostasis of epithelial tissues and organs. Using Xenopus as a model system, we explore the role of intracellular calcium in apical emergence during the development of mucociliary skin epithelium. Our findings reveal that calcium transients precede the apical emergence of Multiciliated cell (MCC) progenitors and are essential for their insertion into the overlying skin epithelium. Furthermore, we demonstrate that phospholipase C (PLC) activity is required for generating calcium transients, which regulate MCC apical emergence via Calmodulin. The PLC/$Ca^{2+}$/Calmodulin axis is necessary for the function of the apical actin network by influencing its stability. Lastly, we show that intracellular calcium regulates apical emergence in distinct basal progenitors. This study advances our understanding of the molecular mechanisms governing apical emergence and highlights the importance of calcium in coordinating cytoskeletal dynamics during epithelial morphogenesis.

Regular addition of new cells and removal of old ones is necessary for epithelial morphogenesis during embryogenesis and the homeostasis of epithelial organs[1]. New epithelial cells can arise from cell divisions within the epithelial sheet[2,3] or from a pool of basally localized progenitors[4–6]. These progenitor cells give rise to new epithelial cells that join the existing epithelial layer through a series of coordinated actions. Initially, the basal progenitors acquire apicobasal polarity[7], intercalate through the basolateral region of epithelial cells[5,6], anchor at the apical junctions of the epithelium and remodel those junctions to join the epithelium[8] and expand their surface area[9,10], to a size accommodating the cell function[11]. Insertion of basally localized progenitors into an overlying epithelium, termed apical emergence, has been shown to contribute to embryogenesis[5,12–14] and tissue homeostasis[4,15–19].

The study of the molecular and cellular mechanisms regulating the apical emergence of basally situated cells is challenging, due to the complexity of the process and the inaccessibility of tissues presenting this behaviour. Mucociliary skin epithelium development of Xenopus embryos has proven to be an invaluable model and has contributed to our understanding of apical emergence[5,6,20–23]. Research on Xenopus mucociliary epithelium development has shown that multiciliated cell (MCC) epithelial insertion is controlled by progenitor cell differentiation[24], apicobasal polarity establishment[7,25], probing of the mechanical environment[8], microtubule accumulation and post-translational modifications[21,26], actin network polymerization[9], and stabilization[27].

The apical actin network has a central role in MCC epithelial insertion. Specifically, during MCC apical emergence, an apical actin network forms a lattice that generates the cell-autonomous forces required for epithelial insertion and apical surface expansion[9,10]. The apical surface of MCCs hosts hundreds of basal bodies and cilia[28]. Thus, tight regulation of apical actin network architecture and dynamics is essential for the apical emergence of MCCs and their subsequent function in the tissue[11]. Nevertheless, our knowledge regarding the molecular regulation of actin network assembly and stabilization during the integration of new cells into an existing epithelium is still limited.

Calcium ($Ca^{2+}$) is a highly versatile intracellular messenger regulating various cellular processes by modulating the activity of $Ca^{2+}$-binding proteins. For example, $Ca^{2+}$ affects actomyosin contractility

Department of Biological Sciences, University of Cyprus, Nicosia, Cyprus. ✉e-mail: nchris06@ucy.ac.cy; skourip@ucy.ac.cy

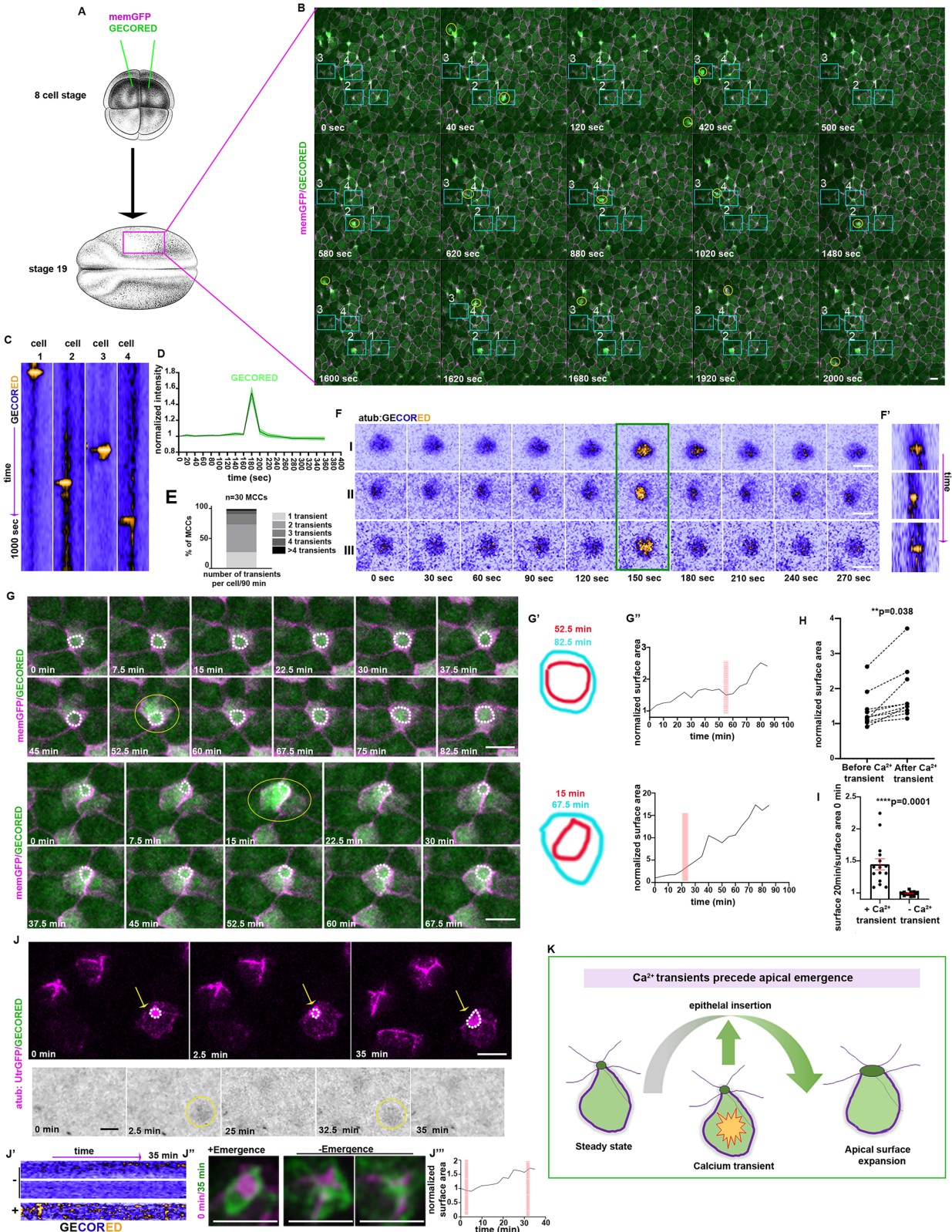

through the regulation of myosin activity, contributing to processes such as cell migration[29] and cell shape changes during embryogenesis[30–32]. In addition to its involvement in actomyosin contractility, it has been demonstrated that $Ca^{2+}$ regulates actin filament formation, F-actin bundling, and severing[33]. The bundling and severing of F-actin is controlled via direct interaction of $Ca^{2+}$ with actin-binding proteins[33]. In contrast, F-actin filament formation is regulated by the $Ca^{2+}$-sensing protein Calmodulin[33]. Upon $Ca^{2+}$ influx, Calmodulin changes conformation and interacts with several cytoskeletal effectors, controlling downstream actin filament formation[34–36]. The role of the $Ca^{2+}$/calmodulin signalling axis during basal progenitor insertion into a polarized epithelium remains to be elucidated.

Here we study the role of intracellular $Ca^{2+}$ in apical emergence during Xenopus mucociliary skin epithelium development. Our work

**Fig. 1 | Calcium transients precede MCC epithelial insertion. A** Diagram of the experimental procedure followed for the generation to time-lapse recordings. **B** Stills from Movie 1 of embryo expressing mem-GFP and GECO-RED. Yellow circles: transient increases in $Ca^{2+}$ levels. **C** Kymographs showing $Ca^{2+}$ levels over time in cells 1–4 marked by rectangles in (**B**). **D** Normalized GECO-RED intensity over time for distinct $Ca^{2+}$ transients. Mean ± SEM. $n = 15$ cells, 2 embryos. **E** Quantification of the number of $Ca^{2+}$ transients per MCC. $n = 60$ MCC, 3 embryos. **F** Stills from representative MCC expressing atub:GECO-RED, 120 s before and after the appearance of a $Ca^{2+}$ transient. **F′** Kymographs of MCCs shown in (**F**). **G** Stills from time-lapse recording showing 2 representative intercalating MCC. Upon the appearance of $Ca^{2+}$ transients, cells expand their surface area. **G′** MCC apical surface area during and after a calcium transient. **G″** Quantification of the surface area over time for MCC shown in (**F**). The red rectangle represents the $Ca^{2+}$ transient.

**H** Quantification of MCC surface area increases in 30 min time window before and after the appearance of $Ca^{2+}$ transient. Two-sided paired student's $t$ test; **$p = 0.038$; $n = 10$ cells, 2 embryos. **I** Quantification of surface area change over a period of 30 min in the presence and absence of a $Ca^{2+}$ transient. Two-sided unpaired student's $t$ test; ****$p < 0.0001$; mean ± SEM $n = 15$ MCCs, 2 embryos. **J** Stills from a time lapse recording following 3 MCC. $Ca^{2+}$transients (yellow circle) appear only in the apically emerging MCC (yellow arrow). **J′** Kymographs for atub:GECO-RED in the three MCCs shown in (**H**). **J″** MIP images of the first and last time points for MCCs shown in (**H**) showing successful apical emergence of the MCC displaying $Ca^{2+}$ transients. **K** Schematic for the temporal relationship between calcium transients and MCC apical emergence. Scale bars: 20 μm. Source data are provided as a Source data file. Xenopus embryo illustrations, ©Natalya Zahn (2022).

reveals that intracellular $Ca^{2+}$ transients precede MCC apical emergence. Subsequently, we show that $Ca^{2+}$ transients are indispensable for MCC insertion in the skin epithelium and subsequent expansion of their surface area in a tissue-autonomous and cell-autonomous manner. Furthermore, we show that PLC activity is necessary for $Ca^{2+}$ transient generation and MCC apical emergence. Additionally, we show that Calmodulin is an effector of intracellular $Ca^{2+}$ during MCC apical emergence, and its activity is crucial in this process. Additionally, we show that the PLC/$Ca^{2+}$/Calmodulin signalling axis regulates apical actin enrichment during MCC apical emergence by influencing the stability of the actin network. Last, we show that intracellular $Ca^{2+}$ is a universal regulator of apical emergence, controlling this process in distinct cell types.

## Results

### Calcium transients precede MCC apical emergence

MCC apical emergence requires the generation of 2D pushing forces from an apical actin network[9]. $Ca^{2+}$-actin interplay is crucial for various cellular processes during development and disease[30,31,37–39]. Thus, to examine a possible involvement of intracellular $Ca^{2+}$ during MCC apical emergence we decided to image $Ca^{2+}$ levels using GECO-RED, a genetically encoded $Ca^{2+}$ indicator[30,40]. The two ventral blastomeres of eight-cell-stage Xenopus embryos were injected with mRNAs encoding membrane-GFP and GECO-RED to target the developing skin epithelium (Fig. 1A). Subsequently, embryos were allowed to develop to stage 19 and imaged during MCC apical emergence (Fig. 1A). Live imaging revealed that intercalating cells frequently displayed transient $Ca^{2+}$ level increases (Fig. 1B, C and Supplementary Movie 1). These $Ca^{2+}$ transients were cell-autonomous and asynchronous, with a duration of 20 s (Fig. 1D), with the majority of cells displaying more than one $Ca^{2+}$ transient (Fig. 1E). To examine if the cells displaying $Ca^{2+}$ transients are MCCs, we cloned GECO-RED under the control of the MCC-specific α-tubulin promoter[5]. Injection of linearized DNA plasmid encoding atub:GECORED revealed that cells displaying $Ca^{2+}$transients are indeed MCCs (Fig. 1F and Supplementary Movie 2). Subsequently, we went on to examine the temporal correlation between $Ca^{2+}$ transients and MCC apical emergence. Thus, we quantified $Ca^{2+}$ levels and apical cell surface area over time in intercalating MCC. This revealed that during MCC apical emergence, $Ca^{2+}$ transients precede MCC apical epithelial insertion and surface area expansion (Fig. 1G–G″, Supplementary Fig. 1, and Supplementary Movie 3). In addition, quantification of MCC apical cell surface area increase over time in the absence or the presence of a $Ca^{2+}$ transient revealed a significant correlation between $Ca^{2+}$ transients and MCCs apical cell surface expansion (Fig. 1H, I). Live imaging of intercalating MCCs before their integration in the overlying skin epithelium revealed that $Ca^{2+}$ transients appear only in MCCs inserting into the skin epithelium (Fig. 1J–J″). Collectively, these results show that $Ca^{2+}$ transients precede MCC apical emergence (Fig. 1K), suggesting that $Ca^{2+}$ transients contribute to the insertion MCC into the overlying skin epithelium.

### Calcium transients are necessary for MCC apical emergence

Intracellular $Ca^{2+}$ transients display strong spatiotemporal correlation with MCC apical emergence. To functionally assess the role of intracellular $Ca^{2+}$ transients in MCC apical emergence, we employed 2-aminoethoxydiphenyl borate (2APB), which blocks $IP_3R$ and other membrane-localized $Ca^{2+}$ channels. MCC start to integrate into the skin epithelium at stage 18 and all MCCs are inserted into the skin epithelium by stage 24[5,21]. Thus, we treated embryos with 2 APB, and we imaged $Ca^{2+}$ levels in intercalating MCCs, revealing that 2APB efficiently blocks $Ca^{2+}$ transients in MCCs (Supplementary Fig. 2A and Supplementary Movie 4), without affecting basal $Ca^{2+}$ levels (Supplementary Fig. 2B, C), in agreement with previous work[31]. Subsequently, we treated embryos with 2APB from stage 15[11,21,41], before initiation of MCC apical emergence and allowed the embryos to progress to stage 24, when MCC apical emergence is completed. In control embryos, MCCs were successfully inserted into the superficial skin epithelium (Fig. 2A, B). In contrast, in 2APB-treated embryos MCCs failed to integrate into the superficial skin epithelium (Fig. 2B). This defect was maintained when embryos were allowed to develop to stage 28, indicating that the phenotype is not due to a delay in the process (Supplementary Fig. 3A–C). 2APB targets ER-mediated $Ca^{2+}$ release, which has been implicated in TJ biogenesis[42,43]. Therefore, we went on to examine if the observed phenotype stems from the disruption of the skin epithelium due to 2APB treatment. For this, we assessed the localization of adherens (β-catenin, E-cadherin) and tight junction markers (ZO-1). This revealed that adherens and tight junctions in 2APB treated embryos were unaffected (Supplementary Fig. 4A–F), indicating that observed phenotype stems from a direct disruption of intracellular $Ca^{2+}$ signalling rather than a generalized impairment of epithelial integrity.

MCC apical emergence is accomplished via distinct independent steps (Supplementary Fig. 3D). MCCs initially intercalate from the deep epidermal layer towards the superficial layer and are positioned between the lateral side of superficial cells[5]. Subsequently, MCCs probe the mechanical properties of the overlying epithelium junctions and actively remodel the overlying junctions to generate high-order vertices of increased tension and anchor at these vertices[8]. Next, MCCs integrate into the skin epithelium and expand their surface area to a size supporting optimal cell function[9–11]. To identify which steps of MCC apical emergence require intracellular $Ca^{2+}$ transients, we injected embryos with atub:UtrGFP, to visualize MCCs, allowed embryos to develop to stage 15 and treated them with 2APB up to stage 24. In both control and 2APB treated embryos MCCs anchor at an overlying high-order vertex (Fig. 2C–E). MCCs in the control embryos successfully insert into the overlying skin epithelium and expand their apical cell surface. MCCs in 2APB treated embryo fails to insert into the epithelium (Fig. 2C–E, Supplementary Fig. 3E, F, and Supplementary Movie 5). Thus, anchoring at high-order vertices was unaffected while insertion of MCCs in the skin epithelium was defective. To examine if $Ca^{2+}$ transients are also necessary for MCC apical cell surface expansion, we

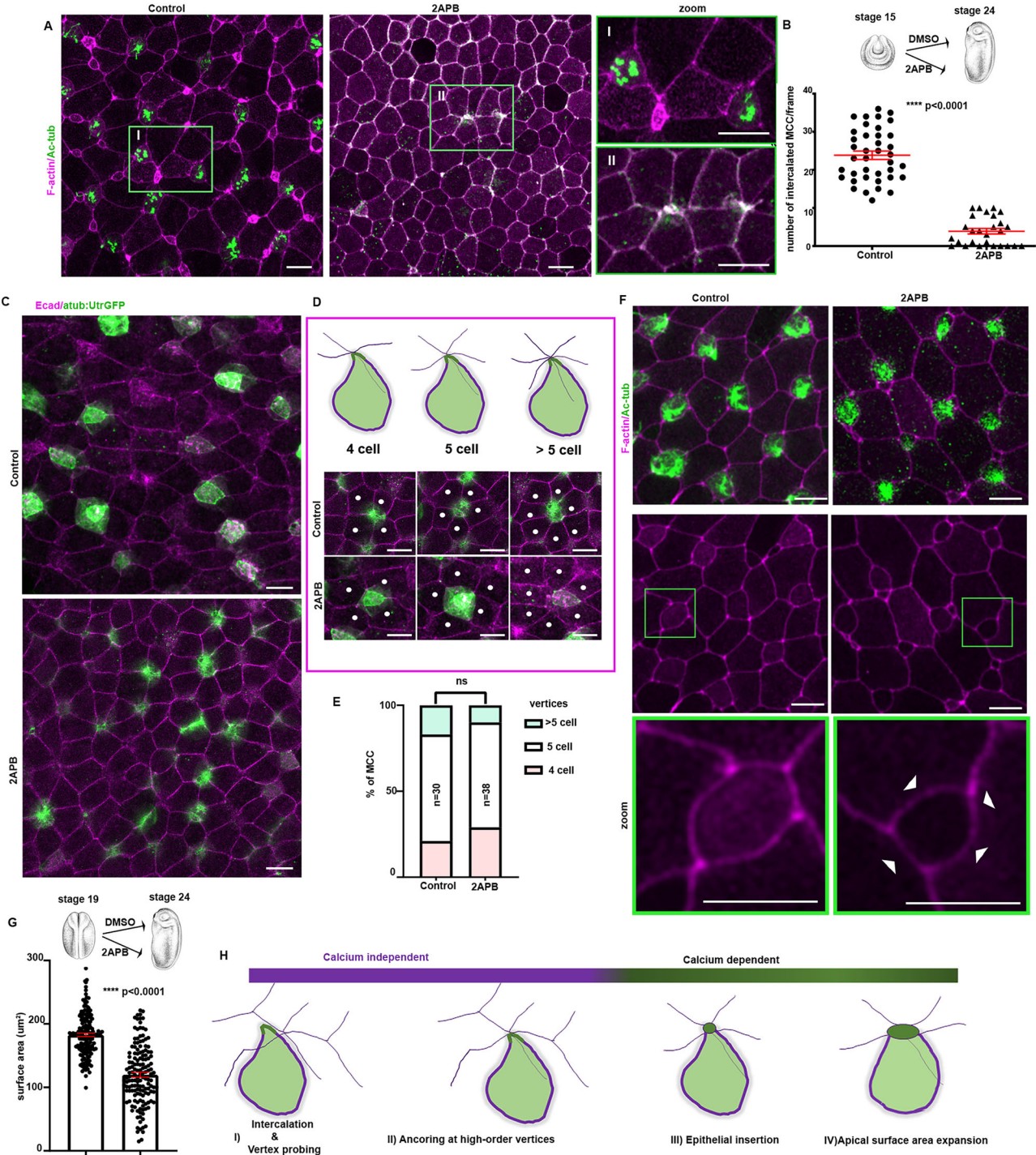

**Fig. 2 | Calcium transients are indispensable for MCC apical emergence.**
**A** Representative images of control and 2APB treated skin epithelium of stage 24 embryo. 2APB treatment started at stage 15. MCCs have successfully inserted into the skin epithelium of control embryos. MCCs apical emergence is defective in 2APB treated embryos, with MCCs failing to acquire an apical surface area (arrows). **B** Quantification of MCC epithelial insertion. Two-sided unpaired student's *t* test; ****$P <$ 0.0001; mean ± SEM. *n* = 39 positions from 24 control embryos and 27 positions from 15 2APB treated embryos. **C** Representative images of control and 2APB treated stage 24 embryos expressing atub:UtrGFP. **D** Representative images of MCCs in control and 2APB treated embryos showing anchoring at high-order

vertices. **E** Quantification of the vertex type occupied by MCCs in 3 control and 3 2APB treated embryos. Two-sided $\chi^2$ test shows no significant statistical differences (ns). **F** Representative images of control and 2APB treated skin epithelium of stage 24 embryo. 2APB treatment started at stage 19. MCCs in 2APB treated embryos insert the epithelium but fail to fully expand their surface area (white arrowheads). **G** Quantification of MCC apical surface area. Two-sided unpaired student's *t* test; ****$p <$ 0.0001; mean ± SEM. *n* = 150 MCC from 5 control and 5 2APB treated. **H** Schematic regarding the role of intracellular $Ca^{2+}$ in the distinct steps of MCCs apical emergence. Scale bars: 20 µm. Source data are provided as a Source data file. Xenopus embryo illustrations, ©Natalya Zahn (2022).

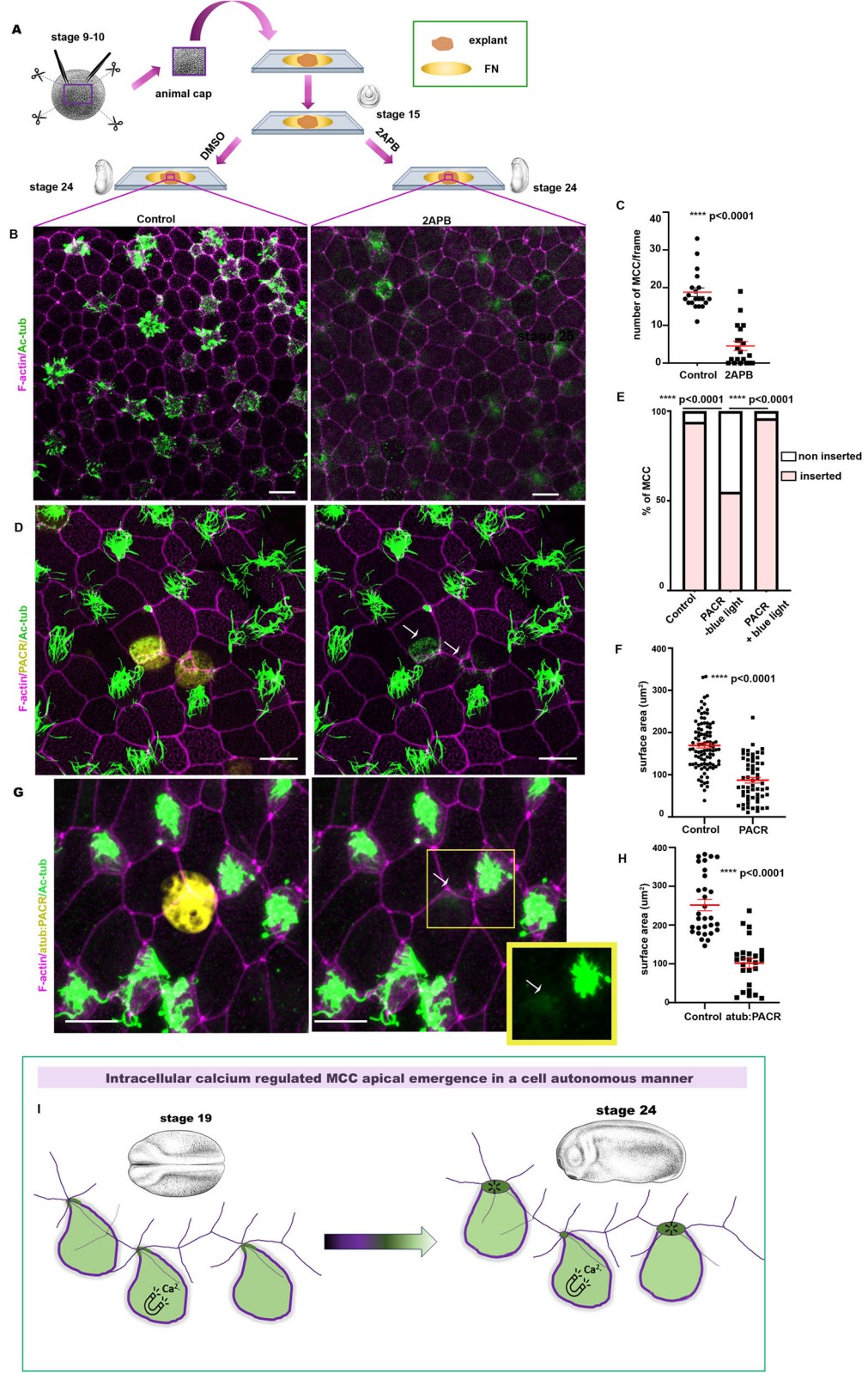

treated embryos with 2APB at the beginning of MCC apical emergence (stage 19) and during MCC surface area expansion (stage 22). These experiments revealed that while MCCs insert into the superficial epithelium in both cases, their surface area is smaller compared to MCCs from control embryos, indicating that apical surface expansion is defective (Fig. 2F, G and Supplementary Fig. 3G–I). The above data reveal that $Ca^{2+}$ transients contribute to specific steps during MCC

apical emergence (Fig. 2H). Specifically, $Ca^{2+}$ transients are dispensable for radial intercalation between the lateral region of superficial epithelial cells, as well as the mechanical probing of apical junctions and the remodelling of these junctions. On the other hand, our data indicate that intracellular $Ca^{2+}$ transients are indispensable for insertion of MCCs into the overlying epithelium and their subsequent apical cell surface expansion (Fig. 2H).

**Fig. 3 | Tissue and cell-autonomous contribution of calcium transients in MCC apical emergence. A** Diagram depicting the experimental approach followed for the generation of animal-cap explants. **B** Representative images of control and 2APB treated animal-caps. Arrows indicate defective epithelial insertion of MCCs in 2APB treated animal-cap. **C** Quantification of MCC apical emergence. Two-sided unpaired student's *t* test; ****$p$ < 0.0001; mean ± SEM. $n$ = 20 regions from 10 control and 2APB treated explants. **D** Representative image of a stage 24 embryo displaying mosaic expression of PACR-GFP in MCCs (arrows). **E** Quantification of apical emergence of control and PACR-expressing MCCs in the presence and absence of blue light (460 nm). Two-sided $\chi^2$ test; ****$p$ < 0.0001. $n$ = 100 control MCCs from 5 different embryos, 108 PACR expressing MCCs from 5 embryos developed in the dark, and 55 PACR expressing MCCs from 4 embryos exposed to blue light. **F** Quantification of the apical cell surface area of control and PACR-expressing MCCs. Two-sided unpaired student's *t* test; ****$p$ < 0.0001; mean ± SEM. $n$ = 97 control and 59 PACR expressing MCCs from 5 different embryos. **G** Representative image of skin epithelium from a stage 24 embryos displaying expression of atub:PACR-GFP (arrow). MCC expressing atub:PACR-GFP fails to successfully integrate into the superficial epithelium (arrow). **H** Quantification of the apical cell surface area of control and atub:PACR-GFP expressing MCCs. Two-sided unpaired student's *t* test; ****$p$ < 0.0001; mean ± SEM. $n$ = 30 control and 26 atub:PACR expressing MCCs from 4 different embryos. **I** Schematic showing the cell-autonomous role of intracellular $Ca^{2+}$ transients in MCC epithelial insertion. Scale bars: 20 μm. Source data are provided as a Source data file. Xenopus embryo illustrations, ©Natalya Zahn (2022).

## Cell-autonomous role of calcium transients

During MCC apical emergence, the skin epithelium receives forces from the underlying extending mesoderm. Mechanical tissue coupling has been shown to affect tissue morphogenesis in different models[44–48]. Specifically, stretching forces generated by mesoderm morphogenesis directly impact the planar cell polarization of MCCs[49,50] as well as the size of MCCs[11]. $Ca^{2+}$ transients have been shown to drive mesoderm convergent extension[51]. To exclude the possibility that mesoderm defects affect mucociliary epithelium development in embryos treated with 2APB, we decided to examine the effect of 2APB treatment in ectoderm (animal cap) explants[52]. These explants differentiate autonomously into a functional mucociliary epithelium and effectively replicate the initial stages of skin epithelium morphogenesis[53]. Animal cap explants were dissected from stage 9 embryos and cultured until sibling embryos reached stage 15 (Fig. 3A), when explants were treated with DMSO and 2APB until sibling embryos reached stage 24 (Fig. 3A). Explants treated with DMSO formed a mucociliary epithelium with MCCs successfully integrated into the superficial epithelial layer as expected (Fig. 3B, C). In contrast, MCC apical emergence was defective in explants treated with 2APB (Fig. 3B, C). The latter indicates that the role of $Ca^{2+}$ transients in MCC apical emergence is tissue autonomous.

During MCC apical emergence MCCs interact with goblet cells and this interaction is essential for insertion of MCCs in the superficial epithelial layer[5,9,53]. The existence of $Ca^{2+}$ transients of low amplitude out of our detection limit in goblet cells cannot be excluded. Thus, treatment of 2APB might lead to inhibition of $Ca^{2+}$ signalling in goblet cells, which might be necessary for junction remodelling in these cells, indirectly affecting MCC apical emergence. To exclude the possibility that the defects observed in MCC apical emergence stem from defects in goblet cells' junction remodelling, we decided to use the genetically encoded $Ca^{2+}$ cage, PACR[54]. PACR combines a photosensitive protein domain, LOV2, and a $Ca^{2+}$ binding domain. When PACR is overexpressed in the absence of blue light, $Ca^{2+}$ is bound to PACR, causing $Ca^{2+}$ to be sequestered from the cytoplasm[54]. Therefore, PACR can be used as a genetically encoded $Ca^{2+}$ chelator. Injection of linear plasmid DNA encoding PACR-GFP led to mosaic expression of PACR in the skin epithelium. Imaging of stage 24 embryos (developed in the dark) revealed that PACR⁺ MCCs displayed defects in epithelial insertion and apical cell surface expansion (Fig. 3D–F and Supplementary Movie 6). These defects were rescued when embryos expressing PACR were exposed to blue light (460 nm) from stage 12.5 to stage 24 (Fig. 3E and Supplementary Fig. 5). To ensure that the defects observed by PACR are not due to ineffective specification, we cloned PACR under the control of the MCC-specific a-tubulin promoter[55]. MCC expressing atub:PACR could not successfully enter the skin epithelium while neighbouring control MCCs successfully entered the epithelium (Fig. 3G, H). Overall, the above data show that $Ca^{2+}$ is necessary during MCC apical emergence in a cell-autonomous manner (Fig. 3I).

## PLC activity is necessary for MCC apical emergence

Inhibition of IP3R-mediated $Ca^{2+}$ efflux upon 2APB treatment leads to defective MCC apical emergence. IP3R-mediate $Ca^{2+}$ release is activated by PLC regulated PIP2 hydrolysis, which leads to the generation of IP3 and DAG[56,57]. Thus, we hypothesized that PLC activity could regulate $Ca^{2+}$ mediated MCC apical emergence. To explore this hypothesis, initially, we sought to monitor PLC activity during MCC apical emergence. For this, we employed a DAG biosensor, to monitor the dynamics of PLC mediated PIP2 hydrolysis[58]. Live imaging of stage 19 embryos expressing a GFP-PKC-γ-C1a revealed that PLC displays pulsed activity during MCC apical emergence (Fig. 4A, B and Supplementary Movie 7). This supports the hypothesis that PLC is involved in $Ca^{2+}$ transient generation during MCC apical emergence. To directly examine the role of PLC for $Ca^{2+}$ transient generation, we first used the PLC inhibitor U73122[59] and imaged intracellular $Ca^{2+}$ levels during MCC apical emergence. Live imaging revealed that PLC activity is necessary for $Ca^{2+}$ transients' generation in MCCs (Fig. 4C, D and Supplementary Movie 8). Next, we went on to examine the role of PLC during MCC apical emergence. We allowed sibling embryos to develop to stage 15 when embryos were treated with DMSO or U73122 until stage 24. Blockage of PLC activity abrogated MCC apical emergence, with MCC failing to enter the superficial epithelial layer or displaying defects in apical cell surface expansion (Fig. 4E, F). U73122 has been reported to have off-target effects. To exclude the possibility that defective MCC apical emergence might stem from the effect of U73122 in other epithelial cell types, we decided to use a well-characterized dominant negative construct for PLC, PH-PLCD1[60,61]. Overexpression of PH-PLCD1 blocks PIP2 hydrolysis by PLC through high-affinity binding to PIP2. To assess the role of PLC specifically in MCCs, we microinjected DNA of a plasmid construct encoding PH-PLCD1 to achieve mosaic expression. PH-PLCD1 resulted in defective MCC apical emergence in the skin epithelium of stage 24 and stage 30 embryos, mimicking the phenotype induced by the pharmacological inhibition of PLC (Fig. 4G–I and Supplementary Fig. 6A, B). In summary, our data show that PLC activity is necessary for $Ca^{2+}$ transient generation in MCCs and is essential for proper MCC apical emergence in a cell-autonomous manner.

## Calmodulin is a downstream effector of calcium signalling

Intracellular $Ca^{2+}$ is a second messenger regulating distinct morphogenetic processes via the regulation of various $Ca^{2+}$-binding proteins[30,62]. One well-characterized $Ca^{2+}$ signalling effector is Calmodulin[63]. Calmodulin (CALM) is composed of four EF-hand domains, each capable of binding a $Ca^{2+}$ ion. Upon Ca2+ binding, calmodulin undergoes a conformational change which affects its interaction with many proteins[64,65]. Previous studies have shown that calmodulin localizes to flagellar radial spokes[66]. In addition, it has been reported that CALM localizes at the centrosomes[67], a structure similar to basal bodies. To examine the localization of calmodulin in the Xenopus embryo, we used GFP-tagged CALM. CALM-GFP displayed centrosomal localization in epithelial cells in agreement with previous

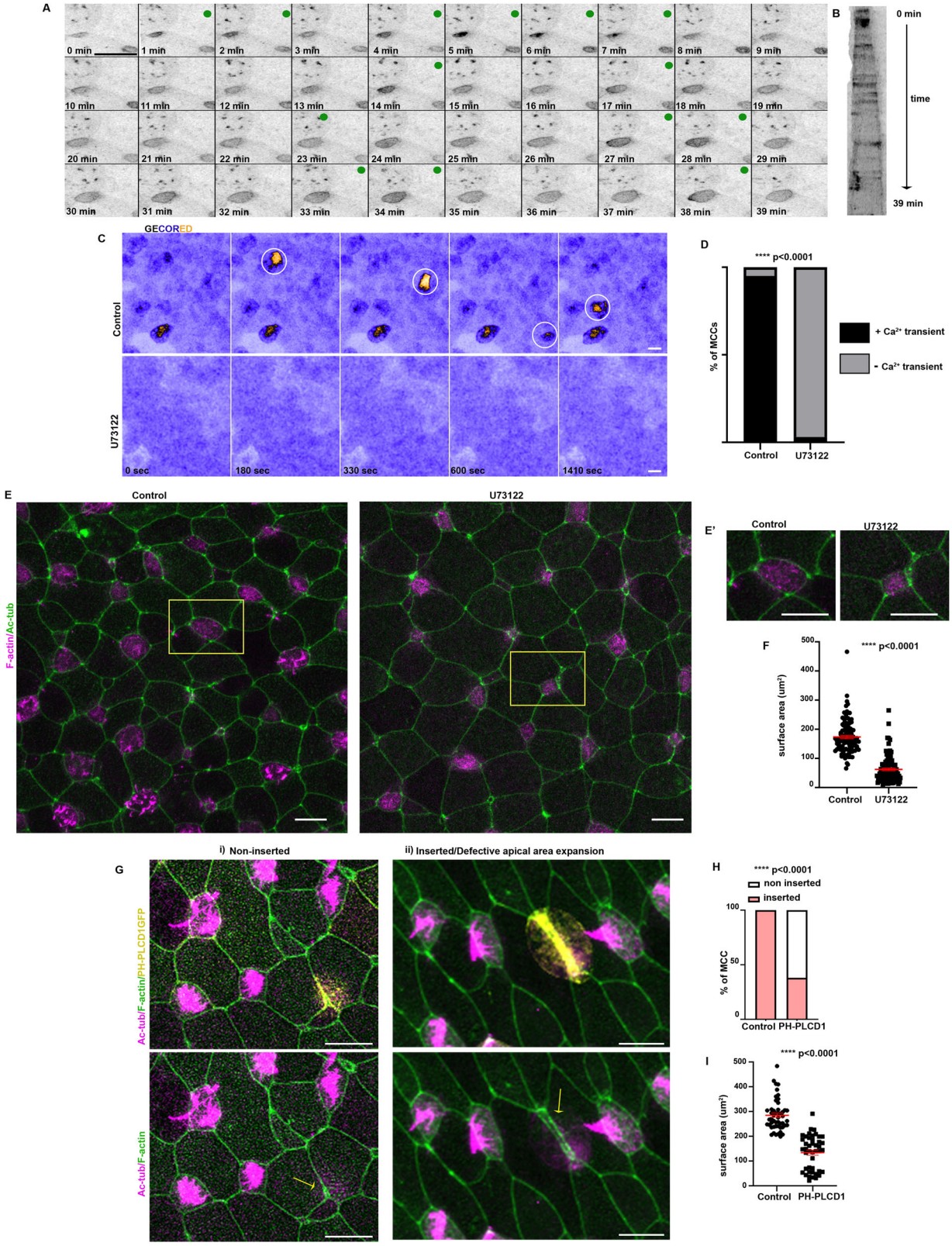

reports (Supplementary Fig. 7A). In MCCs CALM displayed axonemal localization, again in agreement with the reported flagellar radial spoke localization (Supplementary Fig. 7B)[66]. Additionally, at the apical site of MCCs CALM localized at the basal bodies and the apical actin cytoskeleton (Fig. 5A, B). CALM basal body localization was evident before MCC apical emergence and apical basal body docking (Sup-

plementary Fig. 7C). CALM localization suggests a possible role in MCC apical emergence. To examine if CALM activity is necessary for MCC apical emergence, we employed the specific CALM inhibitor, W7. Pharmacological inhibition of CALM resulted in defective MCC apical emergence with MCCs either completely failing to enter the upper epithelial layer or entering the epithelium but failing to expand their

**Fig. 4 | PLC activity is necessary for generation of calcium transients during MCC epithelial insertion. A** Stills from a representative time lapse recording of a stage 19 embryo showing a MCC expressing the PLC activity sensor GFP-PKCγ-C1a. During the insertion into the skin epithelium and the expansion of its surface area, the MCC displays pulsed PLC activity (green dots). **B** Kymograph for the MCC in A depicting the pulsed PLC activity over time. **C** Stills from time lapse recordings showing a region of the skin epithelium of control and PLC-inhibitor treated embryos expressing the $Ca^{2+}$ sensor GECO-RED. Inhibition of PLC suppresses $Ca^{2+}$ transients generation, depicted by white circles in control embryos.
**D** Quantification of $Ca^{2+}$ transients events in MCCs of control and U73122 treated embryos. $n = 60$ MCCs from 5 control and 5 U73122-treated embryos. Two-sided $\chi^2$ test; ****$p < 0.0001$. **E** Representative images of the skin mucociliary epithelium of stage 24 control and PLC inhibitor treated embryos. MCCs fail to enter the superficial epithelial or expand their surface area when PLC activity is impaired. **E′** Zoomed images of MCCs from a control and PLC inhibitor treated embryo (yellow squared in (**E**). **F** Quantification of MCCs apical surface area in control ($n = 100$ MCCs) and PLC-inhibitor treated embryos ($n = 97$ MCCs) from 5 different embryos. Two-sided unpaired student's *t* test; ****$p < 0.0001$; mean ± SEM. **G** Representative images of the skin epithelium of stage 24 embryos expressing the PLC dominant negative construct PH-PLCD1. Expression of PLCD1 results in defective MCC apical emergence (yellow arrows). **H** Quantification of successful MCC epithelial insertion of control ($n = 150$ MCCs) and PH-PLCD1 ($n = 116$ MCCs) expressing MCCs, from 5 embryos. Two-sided $\chi^2$ test; ****$p < 0.0001$. **I** Quantification of apical surface area of control and PH-PLCD1 expressing MCCs that have inserted into the superficial epithelium. $n = 50$ control and 44 PH-PLCD1 expressing MCCs from 5 embryos. Two-sided unpaired student's *t* test; ****$p < 0.0001$; mean ± SEM. Scale bars 20 μm. Source data are provided as a Source data file.

apical surface area (Fig. 5C, D). To examine the role of CALM specifically in MCCs we used a CALM dominant negative construct (CALM1234) that is unable to bind to $Ca^{2+}$ due to mutations in all four EF-hand domains (D20A, D56 A, D93A, and D129A)[68,69]. We injected embryos with plasmid DNA encoding WT-CALM or CALM1234 to achieve mosaic expression. Subsequently, we fixed embryos at stage 32. MCCs expressing the wild-type CALM successfully integrated in the superficial skin epithelium, similarly to control MCCs (Fig. 5E, F). On the other hand, MCCs expressing CALM1234 displayed defects in MCC apical emergence (Fig. 5E–G). Thus, our data indicate that $Ca^{2+}$-bound calmodulin regulates MCC epithelial insertion and apical cell surface expansion in a cell-autonomous manner (Fig. 3I).

### PLC/$Ca^{2+}$/Calmodulin signalling controls actin organization

MCC apical emergence is governed by an apical actin network generating 2D planar pushing forces[9,10]. To examine the influence of intracellular $Ca^{2+}$ in MCC apical network formation during apical emergence, we performed live imaging of stage 21 embryos expressing the $Ca^{2+}$ sensor GECO-RED with Utr-GFP, allowing visualization of F-actin. Live imaging revealed that $Ca^{2+}$ transients precede a transient enrichment of apical actin, which is followed by apical surface area increase (Fig. 6A–C and Supplementary Movie 9). This suggests a possible functional role of $Ca^{2+}$ transients in the regulation of the apical F-actin network in MCCs. To examine if intracellular $Ca^{2+}$ increase is sufficient to induce apical actin enrichment, we decided to pharmacologically elevate intracellular $Ca^{2+}$ using thapsigargin (THA)[30]. Live imaging of embryos expressing atub:GECORED/atub:UtrGFP immediately after the addition of THA to the culture medium revealed that an increase of intracellular $Ca^{2+}$ led to the enrichment of the F-actin at the apical region of MCCs (Fig. 6D, E and Supplementary Movie 10). This further indicates that intracellular $Ca^{2+}$ and its downstream effectors might be necessary for the correct formation of the apical actin network during MCC apical emergence. Treatment of embryos with 25 μm 2 APB completely blocks MCC insertion into the skin epithelium, with MCC failing to acquire an apical surface (Fig. 2A, B). To explore the role of $Ca^{2+}$ transients on apical actin network formation, we treated embryos from stage 15 with 12.5 uM 2APB. Embryos were fixed and analysed at stage 24. Treatment of embryos with 12.5 uM of 2APB did not completely block MCC apical emergence but significantly affected MCC apical cell surface area expansion (Fig. 6F, G). In addition, the apical actin network formation was defective in MCCs of embryos treated with 2APB, as evident by the reduction in apical actin intensity (Fig. 6F, H). In agreement with the above, apical actin network was also impaired in MCCs expressing the genetically encoded $Ca^{2+}$ chelator PACR (Fig. 6I–K). Subsequently, we examine the impact of PLC and CALM activity in MCC apical actin network. Embryos were treated from stage 15 with either U73122 or W7 and allowed to develop to stage 24. Immunofluorescence analysis revealed that PLC and CALM inhibition lead to a reduction of apical actin enrichment during MCC apical emergence (Fig. 6L–O). Overall, these data strongly indicate that the

PLC/$Ca^{2+}$/CALM signalling axis regulates MCC apical emergence by regulating the establishment of the apical actin network.

### Calcium regulates MCC apical actin network stability

Apical actin network and MCC apical emergence are influenced via multiple pathways and processes. Specifically, apicobasal polarity establishment[7,25], RhoA activity[10], centriole amplification[26] and actin network polymerization[9] and stabilization[27] are involved in MCC apical emergence. Therefore, we went on to explore the role of $Ca^{2+}$ signalling in the above processes. To examine the effect of $Ca^{2+}$ signalling on apicobasal polarization, we assessed PAR3 localization in MCC of control and 2APB treated embryos. Par3-GFP apical enrichment was unaffected by 2APB treatment (Supplementary Fig. 8A), indicating that MCC apicobasal polarity is not affected by the disruption of $Ca^{2+}$ transients. In agreement, basal bodies apical localization in MCC of control and 2APB treated sibling embryos before MCC epithelial insertion (Supplementary Fig. 8B) was unaffected. These data show that MCC apicobasal polarization does not require $Ca^{2+}$ signalling.

RhoA activity is necessary for the generation of 2D pushing forces during MCC apical cell surface expansion[10]. To examine if $Ca^{2+}$ transients affect RhoA activity, we used the RhoA activity sensor rGBD[70]. During MCC intercalation active RhoA localizes at the basal bodies[70]. Active RhoA localization and basal body docking were not affected in embryos treated with 12.5 μm of 2APB, even though these cells displayed defects in apical cell surface area expansion (Figs. 7A–C and 6G). Thus, Rhoa activation during MCC apical emergence is $Ca^{2+}$ independent.

Abrogation of centriole amplification and a lower number of basal bodies results in delayed MCC apical emergence[26]. Thus, we decided to examine the effect of intracellular $Ca^{2+}$ on centriole amplification. The total number of basal bodies was lower in 2APB-treated embryos (Fig. 7C, D). However, MCCs in these embryos have a smaller apical surface area (Figs. 6G and 7A). The number of basal bodies is scaled to the apical surface area through a mechanosensitive mechanism and a minimum apical surface area can be achieved independent of centriole amplification[11]. Importantly, the number of basal bodies displayed a high correlation with the size of the apical cell surface area both in control and 2APB-treated embryos (Fig. 7D). Thus, our data show that the defects in MCCs apical emergence upon inhibition of $Ca^{2+}$ transients do not stem defective centriole amplification. This is in agreement with published work showing that inhibition of centriole amplification and defective basal bodies apical docking does not affect MCC apical emergence[11,71–73].

Then we went on to examine how intracellular $Ca^{2+}$ affects the apical actin network during MCC apical emergence. For this, we acquired high-resolution images of the apical actin cytoskeleton in control MCCs and MCCs treated with 12.5 uM 2APB. The enrichment of the apical actin network was defective in MCCs of embryos treated with 2APB (Supplementary Fig. 9A, B). In contrast, the overall architecture of the network did not display major defects (Supplementary

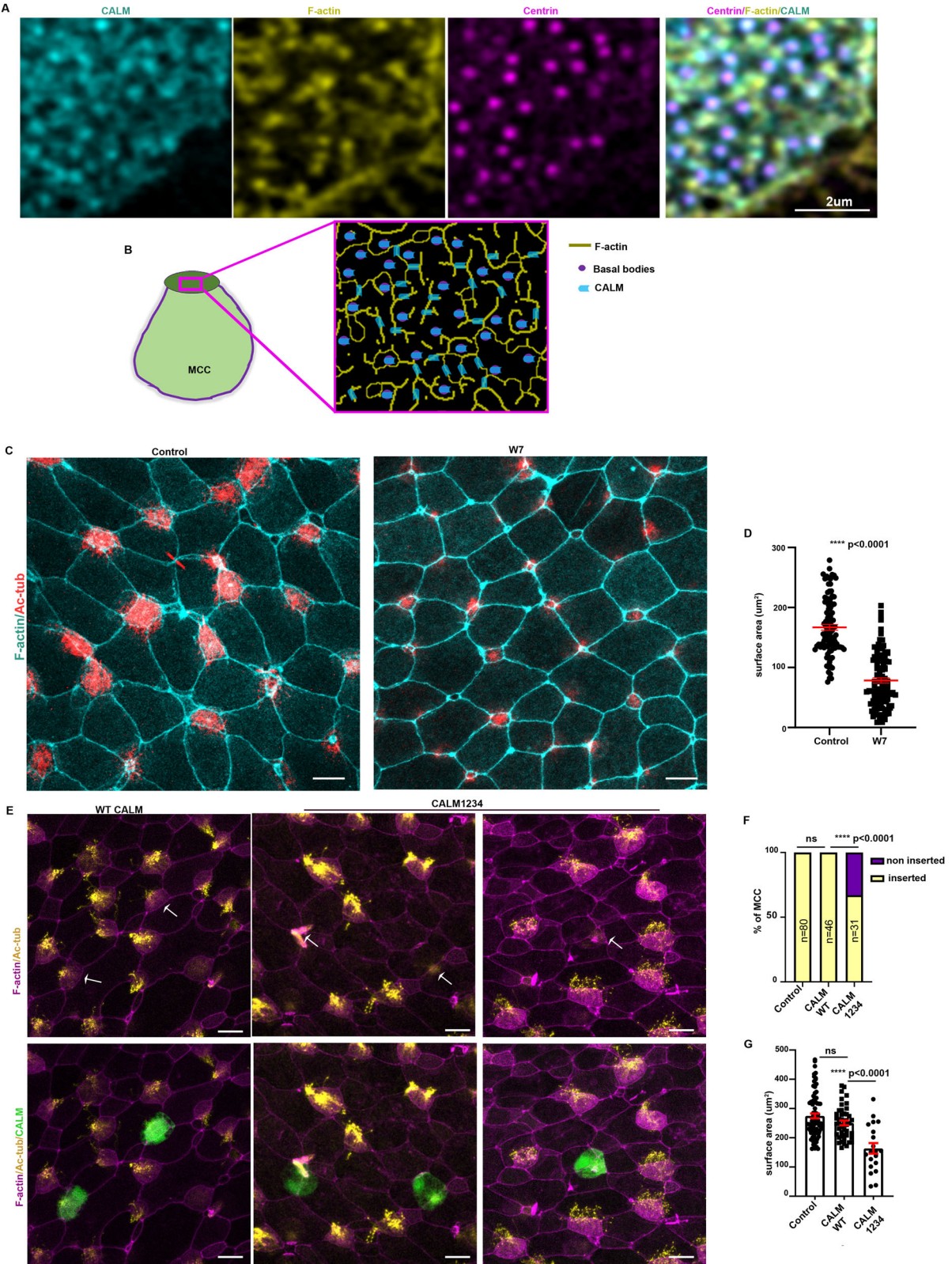

Fig. 9C, D and Fig. 7F–H). This led us to hypothesize that the initial establishment of the actin network can take place in the absence of $Ca^{2+}$ transients, but its maintenance depends on $Ca^{2+}$ transient generation. The F-actin network is a dynamic structure that is constantly assembled and disassembled from G-actin monomers. Since apical actin enrichment is severely affected upon abrogation of PLC/$Ca^{2+}$/ CALM signalling, while the F-actin network architecture is not affected,

we postulated that this signalling axis restricts the pace of actin disassembly. To explore this possibility, we employed Latrunculin A (LatA), which binds to G-actin in a stoichiometric 1:1 ratio and limits actin polymerization[74]. The rate of F-actin network disassembly can be assessed by exposing MCCs to a low concentration of LatA for a predetermined period and monitoring the effects on the apical actin network. We administered a suboptimal concentration of 2APB

**Fig. 5 | Calmodulin regulates MCC apical emergence. A** Representative MIP image of the apical surface area of a MCC expressing calmodulin-GFP. Calmodulin displays colocalization with the basal bodies and the apical actin network. **B** Schematic depicting the localization of calmodulin in a MCC in relation to the basal bodies and the apical actin network. **C** Representative images of the skin epithelium from stage 24 control and calmodulin inhibitor-treated embryos. MCC fail to fully integrate into the superficial skin epithelium when calmodulin activity is blocked, as evident by their small apical surface area. **D** Quantification of the apical surface area of MCCs from control ($n = 100$ MCCs) and calmodulin inhibitor-treated embryos ($n = 100$ MCCs), 5 different embryos. Two-sided unpaired student's $t$ test; ****$p < 0.0001$; mean ± SEM. **E** Representative images of the skin neuroepithelium from embryos expressing wild-type calmodulin or $Ca^{2+}$ binding deficient calmodulin mutant (CALM1234). Expression of CALM1234 results in defective MCC apical emergence (green arrows) while expression of WT calmodulin does not affect MCC apical emergence (white arrows). **F** Quantification of MCC apical emergence upon expression of WT and $Ca^{2+}$ binding deficient mutant calmodulin. Two-sided $\chi^2$ test; ****$p < 0.0001$. **G** Quantification of the apical surface area of MCCs expressing WT and $Ca^{2+}$ binding deficient mutant calmodulin. $N = 80$ control MCCs, 46 MCCs expressing WT calmodulin and 20 MCCs expressing mutant calmodulin from 4 embryos injected with WT calmodulin DNA and 4 embryos injected with mutant CALM1234 DNA. Two-sided unpaired student's $t$ test; ****$p < 0.0001$; mean ± SEM. Scale bars: **A**: 2 µm, **D, F**: 20 µm. Source data are provided as a Source data file.

(12.5 uM) for this experiment to ensure apical emergence of MCCs. Next, we exposed the control and 2APB treated sibling embryos to 2 uM LatA or DMSO for 10 min (Fig. 7E). The apical actin network in MCCs was examined via segmentation of the apical actin network. Both the normalized apical actin network length and normalized number of junctions in the actin network were severely affected only in embryos treated with 2APB+Lat (Fig. 7F–H). This experiment suggests that the apical actin network stability is drastically reduced upon inhibition of $Ca^{2+}$ signalling. To confirm this, we employed FRAP to assess apical actin network dynamics in MCCs from stage 22 control and 2APB-treated embryos[10]. Indeed, the fluorescent recovery of mkate-Actin in 2APB-treated MCCs was faster than that of control MCCs, demonstrating that the apical actin network is destabilized upon 2APB treatment (Supplementary Fig. 9E, F and Movie 12). Thus, defects in MCC apical emergence upon abrogation of normal intracellular $Ca^{2+}$ homeostasis stem from destabilization of the apical actin network (Fig. 7I).

### Calcium controls the apical emergence of distinct cell types

At late neurula stages, the outer skin epithelium of Xenopus embryos is composed exclusively of goblet cells[5]. The mature outer skin epithelium of tadpole stage embryos is composed of 4 different cell types. Goblet cells (GCs), MCCs, Ionocytes (ISCs), and Small Secretory Cells (SSCs)[20,23,75]. The different cell types in the mature skin epithelium can be identified by a combination of staining for Acetylated-Tubulin, F-actin, and PNA-lectin (Fig. 8 A–C). Similar to MCCs, ISCs, and SSCs insert into the superficial skin epithelium through apical emergence of basally specified progenitors in distinct and partially temporally discrete waves (Fig. 8A)[20,22,23]. ISCs apical emergence occurs concomitantly with MCCs, with the process being slower and completed at stage 26[21]. SSCs apical emergence is initiated much later, after stage 25 and is completed by stage 32[23]. Initially, we decided to examine whether $Ca^{2+}$ transients are also present in other cell types during apical emergence. Thus, we performed live imaging of stage 24 embryos, when apical emergence of MCCs is completed and apical emergence of ISCs is ongoing. This revealed spontaneous $Ca^{2+}$ transients in cells inserting into the superficial epithelium and are in contact with MCCs, marked by atub:UtrGFP (Supplementary Movie 11). It is well documented that most ISCs develop in contact with MCCs in the skin epithelium[76]. Therefore, our data show that ISCs also display $Ca^{2+}$ transients during their insertion into the superficial epithelium. To examine if $Ca^{2+}$ transients are necessary for ISCs apical emergence, we treated stage 19 embryos with 2APB and allowed the embryos to develop to stage 26 when ISCs apical emergence is completed (Fig. 8D). We then stained the embryos for F-actin, Acetylated-tubulin, and PNA-lectin to reveal the different cell types on the skin epithelium (Fig. 8A–C). This showed that inhibition of $Ca^{2+}$ transients results in defective MCC apical emergence. Additionally, ISCs numbers are drastically reduced in 2apb-treated embryos and apically emerged ISCs have a smaller apical surface area compared to controls (Fig. 8E–G). These data indicate that intracellular $Ca^{2+}$ transients are necessary for successful ISCs apical emergence and suggest that $Ca^{2+}$ might be a universal regulator of apical emergence. To examine the latter, we decided to examine the effect of 2APB treatment on SSCs apical emergence. To specifically examine SSCs apical emergence, we allowed embryos to develop to stage 27 when MCCs and ISCs apical emergence is completed, and then we treated embryos with 2APB until stage 33 when SSCs are integrated into the skin epithelium (Fig. 8H). In control stage 33 embryos, the skin epithelium consisted of goblet cells, MCCs, ISCs, and SSCs. In contrast, the skin epithelium of 2APB was devoid of SSCs, while MCCs and ISCs apical emergence was not affected (Fig. 8I, J). These data show that intracellular $Ca^{2+}$ regulates SSCs apical emergence and further support our conclusion that $Ca^{2+}$ is a key regulator of apical emergence in distinct cell types, and the requirement for calcium transients coincides with the temporally distinct waves of apical emergence of the various cell types of the Xenopus epidermis.

## Discussion

Insertion of basally located progenitors into an existing epithelial layer is essential for tissue morphogenesis and homeostasis. Basal stem cell integration into an epithelial layer necessary for tissue homeostasis has been documented in the mammalian airway epithelium[4], olfactory epithelium[15], cornea[18], prostate[16], and the Drosophila midgut[17,19]. Additionally, this morphogenetic behaviour has been reported during embryogenesis. Single-cell or multicellular apical emergence contributes to endoderm and node formation in mammals[12–14], and mucociliary skin epithelium development in Xenopus[5,6]. In this work, we focused on the role of intracellular $Ca^{2+}$ transients in apical emergence of basally localized progenitors during Xenopus mucociliary skin epithelium development.

Our data show that intracellular $Ca^{2+}$ transients precede MCC apical emergence and are necessary for MCC apical emergence. These $Ca^{2+}$ transients are generated through PLC activity, with calmodulin acting as an effector of intracellular $Ca^{2+}$ during MCCs cell integration. Mechanistically, our work reveals that the PLC/$Ca^{2+}$/Calmodulin signalling axis regulates MCC epithelial insertion by contributing to apical actin cytoskeleton network stability. We propose that defects in the PLC/$Ca^{2+}$/Calmodulin signaling pathway result in inadequate stabilization of the apical actin network, causing its disruption and impairing the apical actin-generated pushing forces essential for MCCs apical emergence. $Ca^{2+}$/Calmodulin regulate actin network architecture and dynamics via interaction with direct interaction with several actin binding proteins[33] or via the activation of downstream effectors. Here we show that calmodulin becomes associated with the basal bodies and the apical actin network in MCCs. This localization of Calmodulin is consistent with a role in the regulation of actin network stability. Future work should focus on the identification of Calmodulin binding partners and downstream effectors regulating actin network stabilization during apical emergence.

Short-lived $Ca^{2+}$ transients regulate apical emergence, a process occurring over the course of hours. It has been reported that $Ca^{2+}$

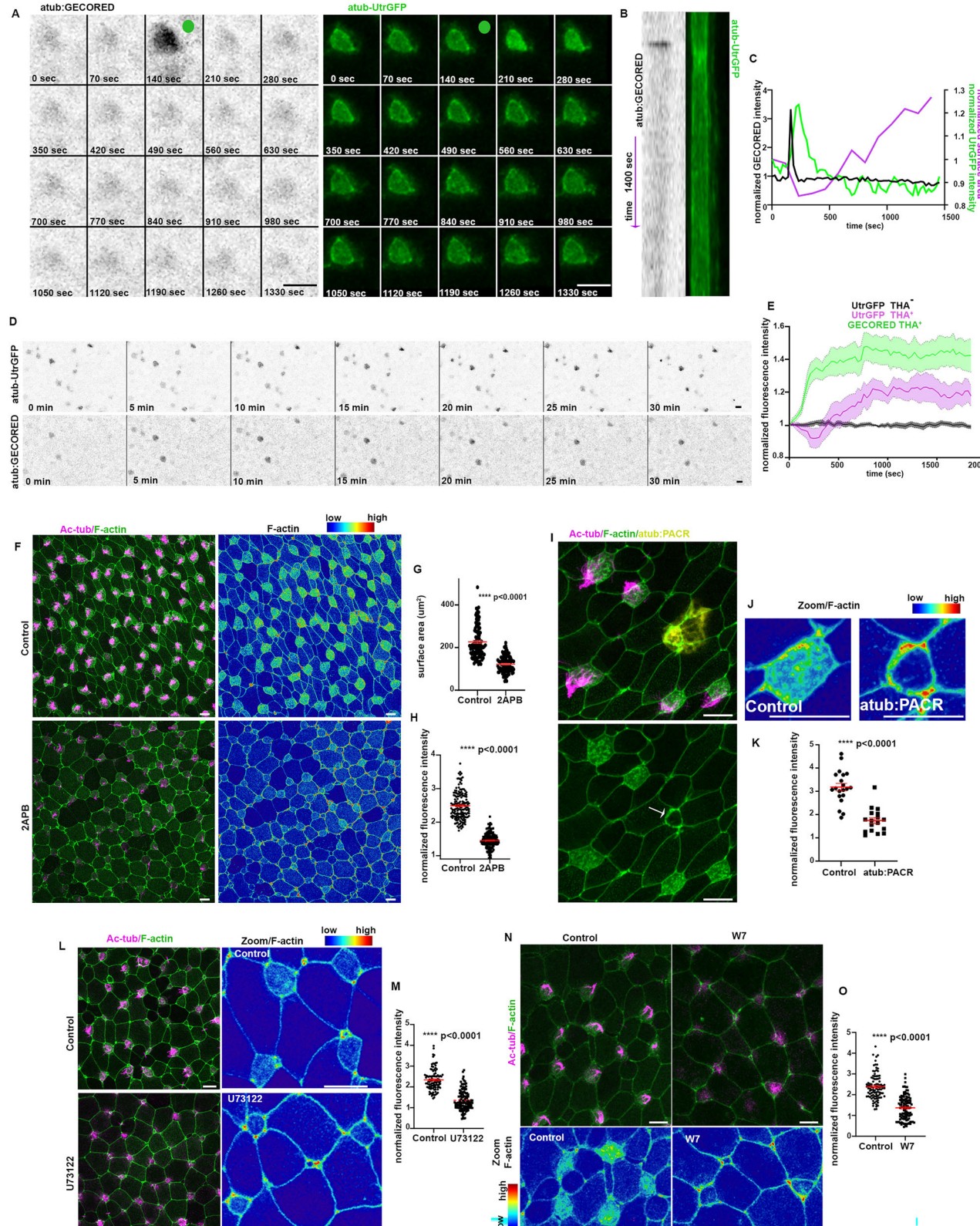

transients can be translated into long-term cellular behaviors during neuronal branching. Specifically, transient Ca²⁺ elevation in neurons activate downstream effectors, including calmodulin, promoting the reorganization of the actin cytoskeleton necessary for the formation and stabilization of new branches[34]. Therefore, transient Ca²⁺ signals can be converted into structural changes that persist for several hours. Similar to neuronal branching Ca²⁺ transients during apical emergence

might lead to activation of downstream effectors that can translate the initial stimulus into persistent cytoskeletal rearrangements. Specifically, during apical emergence, activation of calmodulin during a Ca²⁺ transient can lead to the activation of downstream actin regulators promoting persistent actin network stabilization.

Ca²⁺ transients during MCC apical emergence require PLC activity for their generation. This suggests that PIP2 concentration in MCCs will

**Fig. 6 | PLC/Ca²⁺/Calmodulin signalling governs apical actin cytoskeleton organization in MCC. A** Stills from a time-lapse recording of a MCC from a stage 21 embryo. Green dot: Ca²⁺ transient. **B** Kymograph for MCC shown in (**A**). **C** Quantification of GECO-RED intensity, Utr-GFP intensity and surface area over time for MCC shown in (**A**). **D** Stills from a time lapse recording of a stage 21 embryo treated with thapsigargin. **E** Quantification of GECORED and UtrGFP fluorescent intensity over time for 10 MCCs from (**D**) and UtrGFP intensity over time from 10 control MCCs. Data are presented as mean ± SEM. **F** Representative images from control and 2APB treated embryos. **G** Quantification of the apical surface area of MCCs in control embryos and embryos treated with 12.5 uM 2APB. Two-sided unpaired student's *t* test; ***$p < 0.0001$; mean ± SEM. $n = 100$ MCCs from 5 embryos. **H** Quantification of apical actin enrichment in MCCs from control and 2APB treated embryos. Two-sided unpaired student's *t* test; ****$p < 0.0001$; mean ± SEM $n = 160$ MCCs; 5 embryos. **I** Representative images showing defective apical actin enrichment in atub:PACR-GFP⁺ MCC (white arrow). **J** Fluorescent intensity color coded images of a control and PACR⁺ MCC. **K** Quantification of apical actin enrichment in control and atub:PACR-GFP⁺ MCCs. Two-sided unpaired student's *t* test; ****$p < 0.0001$; mean ± SEM $n = 20$ control and 17 atub:PACR expressing MCCs from 5 embryos. **L** Representative images of control and PLC inhibitor (U73122) treated embryos. **M** Quantification of apical actin enrichment in MCCs from control and U73122 treated embryos. Two-sided unpaired student's *t* test; ****$p < 0.0001$; mean ± SEM. n = 100 MCCs from 5 control embryos and 120 MCCs from 5 U73122 treated embryos. **N** Representative images of the skin epithelium of control and calmodulin inhibitor (W7) treated embryos. **O** Quantification of apical actin enrichment in 100 MCCs from 5 control and 110 MCCs from 5 W7 treated embryos. Two-sided unpaired student's *t* test; ****$p < 0.0001$; mean ± SEM. Scale bars: 20 µm. Source data are provided as a Source data file.

be important in this process, since PIP2 hydrolysis by PLC generates IP3, necessary for Ca²⁺ release from ER stores[56,57]. In addition to IP3, PIP2 hydrolysis results in the generation of DAG. In this work, we don't study the input of DAG in MCC apical emergence. However, it is possible that DAG production could regulate MCC apical emergence or other processes during mucociliary epithelium development through the activation of conventional or novel PKC[77].

Our findings reveal that Ca²⁺ transients regulate apical emergence of distinct basally located progenitors during Xenopus mucociliary skin epithelium development. Specifically, we present data showing that intracellular Ca²⁺ transients regulate not only MCC apical emergence but also the apical emergence of ISCs and SSCs. These data indicate that Ca²⁺ is a universal regulator of this process. Addition of basally located progenitors into an existing epithelium has been documented during tissue morphogenesis and homeostasis contributing embryogenesis and adult organ function[4,5,12–19]. Our work suggests that Ca²⁺-mediated apical emergence may be a conserved mechanism regulating the integration of new cells into established epithelial structures across various tissues and organisms.

## Methods
### Xenopus embryos, microinjection, and cloning
Female adult *Xenopus laevis* frogs were induced to ovulate by injection of human chorionic gonadotropin. All experiments were approved by the National Committee for the Protection of Animals Used for Scientific Purposes of the Republic of Cyprus with the license (CY/EXP/PR.L05/2022). Eggs were fertilised in vitro, after acquisition of testes from male frogs, dejellied in 2% cysteine (*pH* 7.8) and subsequently reared in 0.1× Marc's Modified Ringers (MMR). mRNA and microinjection. For microinjections, embryos were placed in a solution of 4% Ficoll in 0.33× MMR and injected using a glass capillary pulled needle, forceps, a Singer Instruments MK1 micromanipulator and Harvard Apparatus pressure injector at the 4-cell stage. After injections, embryos were reared for 1 h in 4% Ficoll in 0.33× MMR and then washed and maintained in 0.1× MMR. Injected embryos were allowed to develop early tailbud stage (stage 19) at 17 °C and imaged live or allowed to develop to the appropriate stage and then fixed in 1× MEMFA for 2 h at room temperature. Capped mRNAs encoding were in vitro transcribed using Message Machine kits (Ambion). The amount of mRNA per 4 nl of microinjection volume was as follows: membrane-GFP, 100 pg; GECO-RED, 150 pg; rGBD, 100 pg; RFP-Centrin:80 pg; mkate-βactin 80 pg. For microinjection of DNA constructs, atub:UtrGFP, atub:GECO-RED, PACR (Addgene #55774), atub:PACR, CALM-GFP (Addgene #47602), CALMWT (Addgene #111499), CALM1234 (Addgene #111518). Par3GFP, GFP-PKC-γ-C1a (Addgene #21205), PH-PLCD1 (Addgene #21179), Cameleon (Addgene #51961) we injected 80 pg of DNA per blastomere. The atub:GECORED and atub:UtrGFP constructs were generated by replacing the UtrophinGFP coding sequence in atub:UtrGFP plasmid with the coding sequences of GECORED and GFP-PACR using the In-fusion cloning kit. Primer

sequences were as follows: atub-FW:TAACTCGAGCCTCTAGAACTA-TAG, atub-RV:GGTTTGGATCAATTCGAATCGATG, GECORED-FW: GAA TTGATCCAAACCATGGTCGACTCATCACGTC, GECORED-RV: TAGAGG CTCGAGTTACTACTTCGCTGTCATCATTTGTAC, GFPPACR-FW: GAAT TGATCCAAACCATGGTGAGCAAGGGCGAGG, GFPPACR-RV: TAGAGG CTCGAGTTAGAGCTCCAGTGCCCCGGAGC.

### Immunofluorescence
Immunofluorescence was performed as previously described[62]. Briefly, embryos were fixed for 2 h at room temperature with MEMFA, permeabilized in PBST (1 × PBS, 0.5% Triton, 1% dimethyl sulfoxide) and blocked for 1 h in 10% donkey serum. For ZO-1 staining embryos were fixed in methanol over-night at −20 and before blocking the embryos were rehydrated with serial dilution of methanol/PBS. Primary antibodies were incubated overnight at 4 °C. We used a primary antibody against acetylated tubulin, β-catenin (1:500, 11279-R021, Sino Biological), E-cadherin (1:100,5D3, DSHB), and ZO-1 (1:200, 21773-1-AP, Proteintech). Embryos were washed in PBST and incubated for 2 h with secondary antibodies at RT, washed several times and post-fixed in 1 × MEMFA. Secondary antibodies used were Alexa Fluor 488 (1:500, Invitrogen), Alexa Fluor 568 (1:500, Invitrogen) Phalloidin and PNA-lectin were incubated together with the secondary antibodies. Phalloidin 647 plus(A30107, 1:500, Invitrogen), Phalloidin 488 (A12379, 1:500, Invitrogen), Lectin 647 (L32469, Invitrogen).

### Ectoderm explants
Animal caps were dissected at stage nine as described previously[52]. Subsequently, the animal caps were transferred to a slide treated with fibronectin (25 µg/ml) and cultured in Danilchik's for *Amy* (DFA) media supplemented with antibiotic/antimycotic until sibling embryos reached stage 24.

### Inhibitor Treatment
Embryos were treated with 2APB (Abcam, ab120124) 25 uM and 12,5 uM), U73122 (Santa Cruz Biotechnology, sc-3574) 2.5 uM and W7 (Tocris, 0369) 50uM. Before treatment with the inhibitors, the vitelline membrane was mechanically removed, and embryos were allowed to recover for at least 1 h. For Lat treatment stage 24 embryos were treated with 10 uM Latrunculin A (Tocris, 3973) for 10 min when embryos were fixed.

### Imaging
Live imaging of Xenopus embryos was performed on a ZEISS LSM 710 confocal microscope with a Plan-Apo 40×, NA 1.1 or a Plan-Apo 25×, NA 0.8 objectives. The ZEISS ZEN software was used during imaging. In order to prevent the coverslip from pressing against the epidermis, embryos were imaged in a customized chamber composed of a thick layer of vacuum grease on a microscope slide. During imaging, embryos were stored at room temperature and mounted in 0.1× MMR. For quantification of basal Ca²⁺ levels, we used the FRET

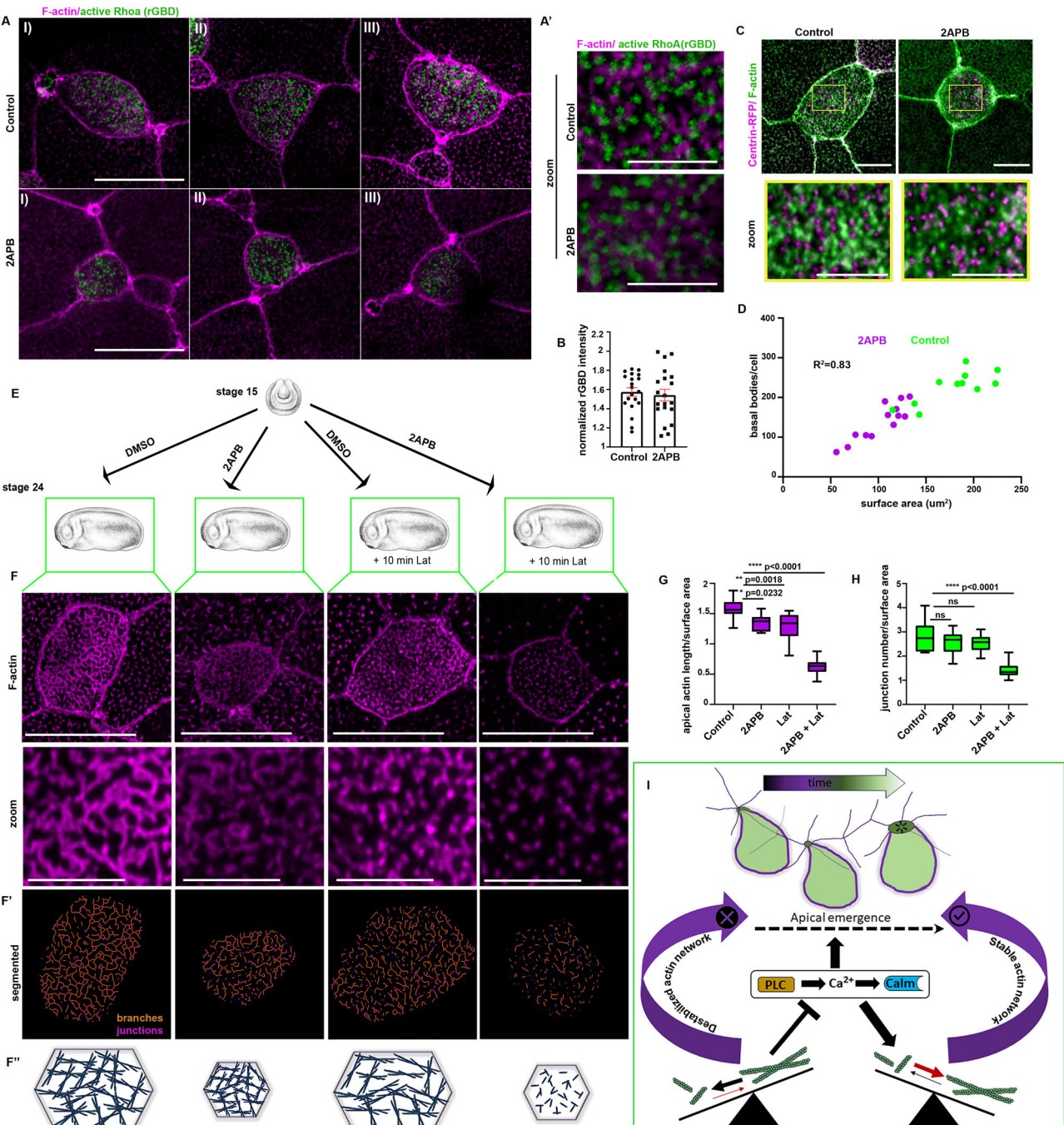

**Fig. 7 | Intracellular calcium controls the stability of the apical actin cytoskeleton in MCC. A** Representative images showing active-RhoA localization in MCCs of control and 2APB treated embryos. **A'** Zoomed images showing that the basal body localization of active-RhoA is not affected in embryos treated with 2APB. **B** Quantification of active-Rhoa localization. Two-sided unpaired student's *t* test; $p = 0.662$; mean ± SEM. $n = 20$ MCCs, from 3 control and 3 2APB treated embryos. **C** Representative images of MCCs expressing Centrin-RFP(basal bodies). **D** Correlation plot between basal body number and apical surface area. $n = 11$ MCCs from 2 control embryos and 13MCCs from 3 2APB treated embryos. Two-sided Pearson correlation test was used, ****$p < 0.0001$. **E** Schematic depicting the experimental design to assess the impact of 2APB treatment on MCC apical actin network stability. **F** Representative images of the apical actin network in control, 2APB treated, Lat treated, and 2APB + Lat treated MCCs. **F'** Segmented images of the apical actin cytoskeleton from the images in (**F**). **F''** Schematics summarizing

the status of MCC apical actin network in different conditions. **G** Quantification of the total apical actin network normalized against the apical surface area. 12.5uM 2APB and 10 min Lat treatment has a minimal effect on the total actin network length. Combination of these treatments results in the destruction of the apical actin network. **H** Quantification of total junction points of the apical actin network normalized against the surface area. The actin network's architecture lost only 12.5 µm 2APB and 10 min Lat treatments are combined. For G and H One-way ANOVA test was used. $n = 10$ MCCs from 3 control, 3 2APB treated, 2 Lat treated embryos, and 15 MCCs from 3 Lat+2APB treated embryos. In box plots, centre lines show median values, box limits represent the upper and lower quartiles, and whiskers show the range of values. **I** Proposed model for the action of PLC/Ca$^{2+}$/ Calmodulin signalling axis during MCCs apical emergence. Scale bars: **A**, **C**: 20 µm. **A'**, **F**: 5 µm. Source data are provided as a Source data file. Xenopus embryo illustrations, ©Natalya Zahn (2022).

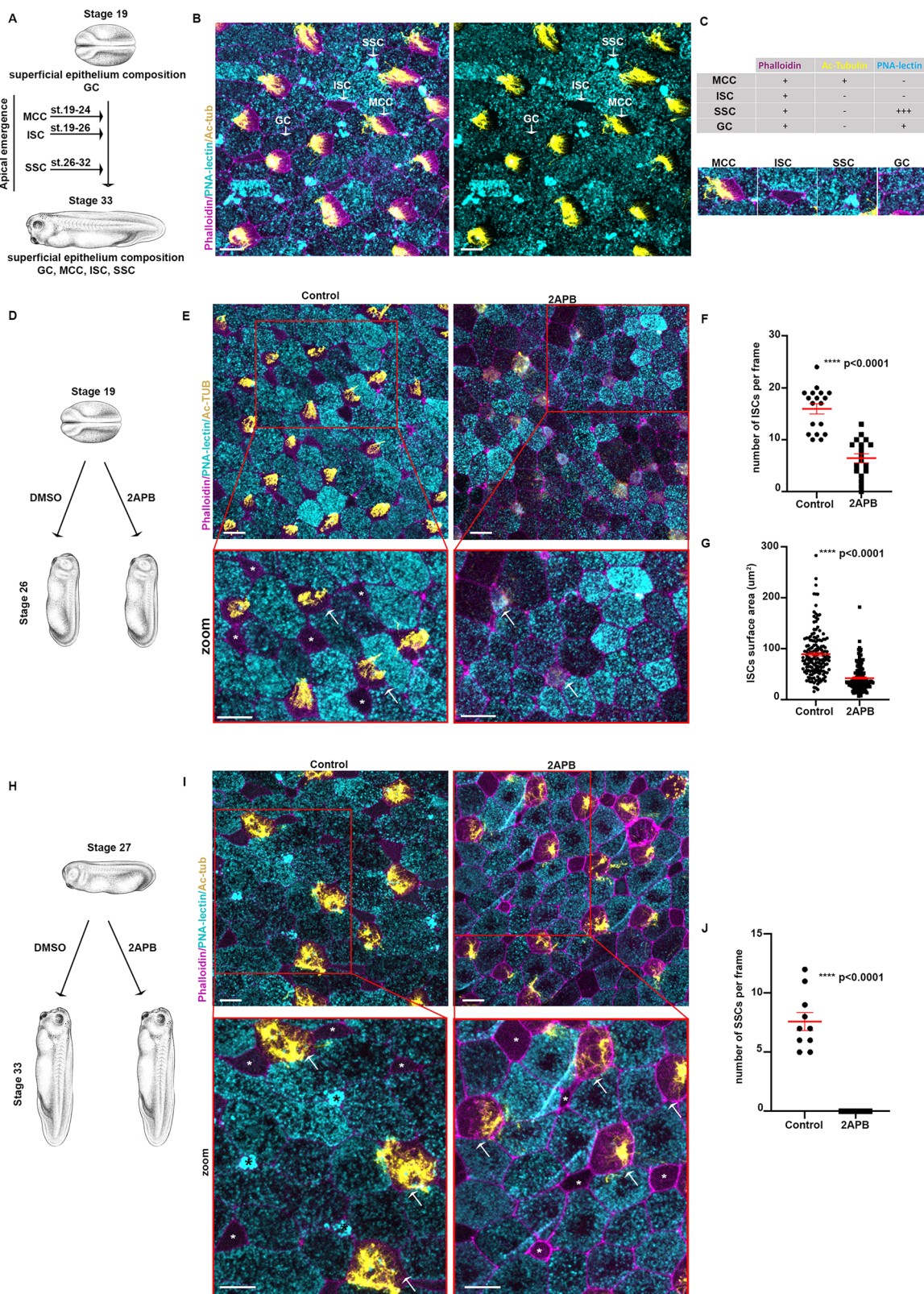

Ca²⁺ biosensor Cameleon[78] in combination with spectral imaging as described before[79]. Briefly, a 458-nm laser was used for excitation (CFP excitation), and emission was recorded using the lambda mode (LSM 710, ZEISS) for spectral imaging (λ scan). Spectral unmixing of the acquired images revealed one emission peak at 478 nm (maximum emission of CFP) and a second one at 527 nm (maximum emission of YFP).

### PACR optogenetic experiments

For our optogenetic experiments, we used a LED bulb (Higrow, GL36B) that emits BLUE light (460 NM), inhibiting Ca²⁺ binding to PACR. To examine the effect of Ca²⁺ chelation upon PACR expression, embryos expressing PACR-GFP were allowed to develop until stage 24 in the dark, by wrapping a 6-well plate in aluminium foil. To examine if exposure to blue light rescues the phenotype induced by PACR

**Fig. 8 | Calcium regulates the apical emergence of distinct basally localized progenitors. A** Schematic depicting the stages of apical emergence for MCCs, ISCs, and SSCs. **B** Representative images of the skin epithelium of a stage 33 embryo. The different cell types of tissue are highlighted. **C** Table and representative images for depicting how F-actin, Ac-tub, and PNA-signals were used for the identification of the different cell types. **D** Schematic for the experimental protocol used to assess the role of intracellular $Ca^{2+}$ transients during ISC apical emergence. **E** Representative images of a control and 2APB treated stage 26 embryos. Arrows: MCCs. Asterisks: ISCs. **F** Quantification of apically emerged ISCs per field of view. Two-sided unpaired student's *t* test; ****$p < 0.0001$; mean ± SEM. $n = 18$ regions

from 9 embryos per condition. **G** Quantification of ISCs apical surface area. Two-sided unpaired student's *t* test; ****$p < 0.0001$; mean ± SEM. $n = 159$ cells from 5 control embryos and 135 cells from 8 2APB treated embryos. **H** Schematic for the experimental protocol used to assess the role of intracellular $Ca^{2+}$ transients during SSC apical emergence. **I** Representative images of a control and 2APB treated embryos. Arrows: MCCs. White Asterisk: ISCs. Black Asterisks: SSCs. **J** Quantification of apically emerged SSCs per field of view. Two-sided unpaired student's *t* test; ****$p < 0.0001$; mean ± SEM. $n = 10$ regions from 5 embryos per condition. Scale bars: 20 μm. Source data are provided as a Source data file. Xenopus embryo illustrations, ©Natalya Zahn (2022).

expression when the embryos were developed in the dark, stage 12.5 embryos were allowed to develop in an optically transparent 6-well plate and exposed to blue light (460 nm) until the embryos reached stage 24.

## Fluorescent recovery after photobleaching

FRAP experiments were performed using a Zeiss LSM 700 confocal. Stage 22 embryos expressing mKate–β-actin were immobilized in silicone grease wells on glass slides. Single MCCs were imaged using a Plan-Apochromat 63×/1.40 Oil DIC M27 objective lens (Zeiss), and a 543 nm laser was used both during acquisition (4.5%) and bleaching (100%). Fluorescent recovery was monitored for 30 frames (1 s time interval), bleach corrected and normalized using Fiji software[80] (http://imagej.net/Analyze_FRAP_movies_with_a_Jython_script).

## Image analysis and figure preparation

All image analysis and quantification were carried out using Fiji software[80]. To analyze apical actin network structure, we followed an automated segmentation protocol previously described[81]. The total length of the apical actin network and the number of junctions in the actin network was normalized by dividing the values of total network length and total junction by the surface area of each MCC. For quantification of MCCs, ISCs, and SSCs insertion into the skin epithelium, we quantified the number of successfully integrated cells in images with the same frame size (212.55 × 212.55 μm). Successful epithelial insertion was considered when cells displayed an apical surface area > 10 μm². For normalization of the surface area, GECO-RED and Utr-GFP signal intensity in live imaging experiments, the apical surface area of MCCs in a given time point was divided by the surface area of the first time point, and the signal intensity of GECO-RED and atub:UtrGFP at a given time point was divided by the signal intensity of the first timepoint. For normalization of the F-actin intensity in MCCs (Fig. 6), the signal intensity of F-actin from a single MCC was divided with the mean F-actin signal intensity of 10 goblet cells from the same field of view. Figures were prepared using Adobe Photoshop. *Xenopus embryo* illustrations used in all Figures are from Xenbase (www.xenbase.org RRID:SCR_003280), Zahn et al. [82], ©Natalya Zahn (2022).

## Statistics

GraphPad Prism 8.0 software was used for all statistical analysis performed. The sample size of the experiments carried out was defined based on previous experimental experience. Quantitative data presented show the mean ± s.e.m., or the total number of datapoints obtained. The statistical tests carried out on the quantitative data obtained are annotated in each legend.

## Reporting summary

Further information on research design is available in the Nature Portfolio Reporting Summary linked to this article.

## Data availability

The authors declare that all data supporting the findings of this study are available within the article and its supplementary information files.

All data can be provided by the corresponding authors upon request. Source data are provided with this paper.

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

## Acknowledgements

We thank Dr Peter Walentek for kindly providing the atub:UtrGFP plasmid. This work was funded by the Cyprus Research and Innovation Foundation (Projects: EXCELLENCE/0421/0003, EXCELLENCE/0524/0002, Small-Scale Infrastructure/1222/0186) under the programme of social cohesion "THALIA 2021-2027", which is co-funded by the European Union.

## Author contributions

Conceptualization: N.C.; Methodology: N.C.; Investigation: N.C.; Writing–original draft: N.C.; Writing–review & editing: N.C., P.A.S.; Funding acquisition: N.C., P.A.S.

## Competing interests

The authors declare no competing interests
