## [Peer Review file · Nature Communications]

Calcium transients regulate the apical emergence of basally located progenitors during *Xenopus* skin development.

Corresponding Author: Professor Paris Skourides

Version 0:

Reviewer comments:

Reviewer #1

(Remarks to the Author)

The manuscript by Chistodoulou and Skourides addresses the mechanism by which basal cells insert into an epithelial sheet. In *Xenopus*, the embryonic epidermis acts as a mucociliary epithelium in which multiciliated cells (MCCs) originate at the basal layer and then must migrate apically and insert into the overlying epithelium. There are many interesting processes here that are not well understood. For example, once the apical tip of the MCC hits the epithelial surface, it must create pushing forces that expand laterally to insert the cell into the epithelium. Additionally, during this process, apical actin creates a "honeycomb" network in which basal bodies dock establishing proper spacing between cilia and defining the correct density of cilia/surface area. Therefore, these MCCs are extremely interesting from their initial specification, apical intercalation, and formation of planar-coordinated beating cilia in an epithelial sheet.

The authors seek to better understand the role of calcium transients in the intercalation of the MCC into the epithelium. The authors identify a phospholipase C- calcium transient – calmodulin pathway that is essential for actin stabilization. They argue that prior to apical emergence of the MCC, calcium transient(s) is/are essential for the stabilization of actin via calmodulin, that can create the pushing forces for lateral expansion of the apical MCC surface.

MAJOR CONCERNS:

1) Calcium transient vs. steady-state calcium concentrations

The authors begin by using a RGECO calcium sensor and detect flashes of fluorescence in MCCs. The main point of Figure 1 is that Ca transients occur before intercalation as measured by apical surface area. I have a number of concerns with this conclusion:

a) the data presented in Fig 1G suggests that the apical surface area increases mildly after a calcium transient. There are three outliers but really the increase is small in the majority of the data before or after calcium transient with most changes from 1.2-1.5 fold with significant overlap before and after the transient (of note the apical area of an intercalating MCC increases dramatically – many fold increase). I also note that plots of the calcium transient vs the surface area of the MCC don't correlate very well. Looking at supplemental figure 1B – panel 1 looks like the increase in surface area occurs 10 min after the transient, in panel 2 concurrent with the expansion, panel 3 10 min after, panel 5 seems to have no change in the acceleration of apical area with two transients. All of these panels have different time windows, and dramatically different fold increases (panel 1, IV 2 fold, while panel 3 is 10 fold). The lack of consistency makes it difficult to correlate the flash with the amount of change in apical area that is essential for their argument of correlation. Additionally, a major concern is the time resolution. It appears that while the movies are taken for long periods of time (30 min) the resolution is low – every 30-40 seconds for transients whose period is not clear. Are there short flashes that occur for 15 seconds (still a substantial flash) that might be missed? How long are these flashes? How many flashes occur per cell? I worry that they may be missing some that might indicate that these flashes are not exclusively occurring prior to the intercalation which could change their interpretation as a signal for epithelial insertion or as pulses that drive waves of actin formation. For example, in the context of PLC/IP3 inhibition, perhaps the role of these molecules is to intensify a flash that is coming from another source. Inhibition with 2APB might still lead to flashes, but these become too short (to detect in their 30 s sampling) and weaker without PLC/IP3 intensification.

b) To test the importance of these transients, the authors seek to inhibit them with 2APB or PACR. While the authors quote a single study stating that 2APB alters calcium transients but not basal intracellular calcium level, they do not measure that in this context. The previous study examined the neural plate and neural plate closure, a very different process than apical intercalation of MCCs. They simply cannot make the argument that MCC intercalation is dependent on calcium transients without confirming that basal calcium levels are unaffected (as was done in the quoted study). We know that actin polymerization depends on calcium and so depletion of basal calcium levels will impair actin which could entirely explain their results. To make the claim that calcium transients drive apical intercalation and not basal calcium levels, the burden of proof is on the authors to demonstrate no change in basal calcium levels.

c) Measuring calcium transients vs basal calcium levels is not a trivial task and requires the use of different calcium sensitive dyes that can detect the different levels with sufficient signal to noise. It is also critical that this be done ratiometrically as the literature is fraught with complications where transient calcium imaging leads to artifactual results because the "load" of calcium dye is not controlled for from cell to cell. This is a major weakness of this paper.

d) no data is presented that 2APB actually eliminates calcium transients. There is simply data showing that the MCCs are not intercalated (an end result). There still could be calcium transients with the 2APB treatment. I agree there is a dramatic effect on MCCs with 2APB treatment but the authors need to confirm that this leads to a loss of calcium transients (and normal basal calcium). Otherwise they simply cannot make a conclusion.

e) PACR data does not resolve this challenge as PACR could very well chelate both basal and transient calcium concentrations. I note the authors do not take advantage of the optical properties of PACR. If the embryos are bathed in blue light, does the MCC phenotype rescue? Can they use the PACR photosensitivity to titrate the chelation to inhibit transient calcium, but not basal calcium?

f) another option would be to load the cells with caged calcium, inhibit with APB, and then release the calcium with a pulse of light to simulate a transient. This would be better than thapsigargin which is stated to be in figure 6d-e but is completely unlabeled in the experiment or lacks a control.

2) Finally, the source of the calcium transients is not clear. The authors suggest that these transients are coming from the ER which is certainly possible but they also could be triggered across the plasma membrane and then amplified by PLC/IP3/ER. There are both voltage gated channels and mechanosensitive channels that could be playing a role at the membrane surface.

Overall, based on these data, I am unconvinced that the authors can claim that calcium transients are critical for MCC intercalation. I don't doubt that calcium is necessary for actin to form the apical actin network and that inhibiting IP3/PLC will reduce the actin, lateral pushing force, and MCC intercalation. However, whether this is due to a transient calcium flash is unclear. Logically, what is the purpose of the 30 sec pulse of calcium? The intercalation of the MCC occurs over 1-2 hours and so what would be the purpose of a 30 second calcium transient – certainly this is not a sustained calcium source necessary for the continued polymerization of actin which must be occurring with basal calcium levels over this time period. As a hypothesis, perhaps at the apical surface prior to intercalation, the calcium transient snaps the balance between actin polymerization and depolymerization favoring polymerization. This would be consistent with their latrunculin A result showing increased depolymerization with the 2APB. However, if that is the case, what is the mechanism? What actin related proteins are "snapped" due to this calcium transient that lead to sustained polymerization (memory of the pulse) over 1-2 hours? This certainly would be very novel and of interest to a broad audience like Nature Communications. But they must prove that basal calcium levels are unaffected, define the sufficiency of the calcium pulse, and the requirement of the calcium pulse over basal levels.

Minor Issues

Fig 1 – In fig 1B- the authors label four cells that they then show a kymogram in Fig 1C. It would be very helpful if the labeling in the top right panel of the Fig 1B indicating the four cells were reproduced in all the other panels at different time points. While I think I can decipher which cell is which, it would be very helpful to have each panel labeled with the four cells.

Supplementary figures and movies – please label each in the figure or in the movie whether they are Supp Fig 1, 2, 3, etc or Movie 1, 2, 3. Movies are labeled as 556027_0_video_9850791_smf468- and so I just guessed that 791 was Movie 1 and 792 was movie 2, etc. For the supp figures, again I just guessed on order – the main figures are nicely labeled with Figure 1, 2, 3, etc in the margin - please do the same with supp. Figures and movies.

Reviewer #2

(Remarks to the Author)

The manuscript by Christodoulou et al. addresses the important question of the role of calcium signaling during radial intercalation. The authors identify calcium transients that precede the ability of cells to apically insert into the epithelium. They go on to use both pharmacological and dominant negative approaches to test the function of intracellular calcium release (2APB and PACR), PLC (U73122 and PLCD1), and Calmodulin (W7 and CALM1234). Finally, they assess the consequence on apical actin during 2APB treatment. Overall they propose that Ca⁺⁺ PLA CALM axis regulates both apical insertion and apical expansion. This paper addresses complex questions and the overall message that Ca⁺⁺ signaling is important to the process is believable. However, much of the data is of poor quality, the presentation is sloppy and there is a

significant amount of over interpretation that drives my enthusiasm significantly down. This is a mature field and the authors have notably attempted to place their work in the broader context but in doing so have created a poorly focused message that is difficult to interpret. That said there are also some interesting findings and with significant work this could be an impactful addition to the field. One of my primary concerns is that they claim that Ca^{++} is important for both insertion and expansion. However, the timing of the experiments make this claim difficult to determine as this is an ongoing process and a delay in insertion will appear as a loss of expansion when really the process is just delayed. To make these claims the experiments would either need to be done more dynamically or at the very least be done past the normal end of the process to capture any delays. As presented, I don't think these separate claims are warranted, but the claim that Ca^{++} is involved in the overall process is justified.

Comments:

Mandatory. The process being studied has been repeatedly termed radial intercalation which has also been called apical emergence being broken down into two steps apical insertion and apical expansion. For the continuity of the field you need to refer to this process as radial intercalation or apical emergence rather than epithelial integration in the title, abstract and throughout the paper. While I don't mean to curb your creativity, rebranding of processes is a scourge on science making it harder for researchers to follow the field.

From the analysis (e.g. 1C,D) it would seem as if there is a single Ca^{++} spike, but the movies make it look like individual cells have multiple spikes. Can you comment on this and whether it takes repeated burst to induce intercalation.

Figure 1G "Quantification of MCC surface area increases in a 30min time window before and after the appearance of calcium transient". Given the stage of this analysis and the length of the measurement one would expect cell size to change so it is hard to assign function to the Ca^{++} transient. Could you compare the size change in cells that do or do not have a Ca^{++} transient?

F' Shows the cell size at two points from the movies presented in F, but it is not clear how those measurements were made as the cells do not look considerably different in the images. This calls into question the data in F'. At the very least show an outline of what is being measured. Better cell boundary imaging would help.

Statistics in 1G should be a paired t test and overall not enough statistic information is provided throughout the manuscript.

The imaging in 1H is so low quality as to be uninformative.

2ABP is not a specific drug making interpretations of its use difficult. Particularly as it also affects TRP channels which are likely involved in this process. I think other experiments validate this finding but this should at least be stated and discussed rather than calling it a specific blocker.

Figure S1C does not have a scale bar and it is hard to see what this is. If it is super low resolution as I suspect then it is not clear what is being quantified. Also the cells in D are not identified. Also quantification doesn't provide n's.

The claims in Figure S2 are problematic. First it is mislabeled making it confusing, but if I understood things correctly I would say that for bCatenin, the staining looks quite different in control versus 2APB, however there are quite a few cells that have intercalated so I am not sure that the drug treatment even worked. Regardless single line scans is not sufficient to make any claims.

Line scans of relative intensity for a single cell are simply not informative. This type of data is repeatedly used throughout the paper.

The description of Figure 2C does not really add any new information to the field and should probably be supplemental, but it should absolutely reference the work that showed these steps first, in particular Sedzinski 2016 and Ventura 2022.

The authors claim that "our data indicate that intracellular calcium transients are indispensable for MCC epithelial integration and apical cell surface expansion (Figure 2H)." But since the 2APB blocks insertion it is hard to say what effect it is having on expansion. Cells that don't insert cannot expand. But also as mentioned above these effects could be delays.

It is not clear why the authors used the Ca^{++} caged optogenetic PACR, as they do no optogenetic experiments. If the cells could intercalate after blue light treatment that would be informative.

Figure 4B is hard to interpret as presented. What step of the process is the cell at, as it looks as if it is already intercalated? The overall cell shape and what looks like the apical surface does not change over the 40 minute movie? It is not clear what they are proposing is happening here or what the consequences are.

U73122 has been reported to have numerous off target effects. While this result is largely validated with the PLCD1 it should still be stated.

Does Thapsigargin promote early apical insertion as would be predicted by your model? This would really strengthen your claims.

"Inhibition of PLC suppresses calcium transients (green circles) generation." The overall GECO-RED signal appears much

higher in the U73122 treated tissue which would confound this interpretation. Please explain.

N's should list embryos tested not just cells. This is true throughout the paper and analyzing 20 MCCs from the same embryos is not the same as analyzing 5 MCCs from 4 embryos.

Many of the figures are sloppy. Perhaps the worst example of this is that the images in 6D are not properly matched with Utr and GECORED. The cropped images only partially match.

"Thus, our data suggest that the defects in MCCs epithelial integration upon inhibition of calcium transients do not stem defective centriole amplification." I am not sure these data totally rule out this possibility but a bigger concern is that it is known that centriole docking facilitates the establishment of the apical actin (Kulkarni et al 2018) and the data in S4B shows that the basal bodies are not docked, which would alter actin.

There is a complete lack of continuity on how experiments were quantified (or not). Data such as 2G should be quantified. Nowhere could I find a description of how insertion was defined for the quantifications of inserted or not. For example does 4H show two cells that fail to insert or one that has inserted (ii).

Others have shown that ac tub marks cells prior to intercalation (Collins 2021), but in 3G there is no acetylated tub in PACR cell. If there is really no acetylated tubulin then the cells would fail to insert based on that.

Figure 5D does not state stage which is essential for interpretation.

It is not clear why the cartoon in 5C has CALM asymmetric to centrin when the example in B looks fairly symmetric.

Figure 6A. I am unclear of the interpretation as Urothrin has been shown to be enriched apically (Yasunaga et al 2022). Why is its enrichment transient and why is it not enriched prior to the Ca^{++} pulse. The way this data is presented is confusing.

What Stage in 6D. Are these cells inserted already? Not sure what the relevance is here to the overall claims of the paper.

Minor:

For figure 1 the text refers to stage 18 but the figure states stage 19.

Why was neuroepithelium used in 4F and what exactly does that mean.

Figure 1B should continue to label the 4 cells that are being tracked.

No scale bar references in the legends.

Centrosomes are comprised of two centrioles which are identical structures to basal bodies (not similar to as the text states). Figure 5C should be labeled centriole not centrosomes.

Reviewer #3

(Remarks to the Author)

In this paper, Christodoulou and Skourides investigate the role of calcium signalling in radial intercalation of multiciliated cells (MCCs) in the *Xenopus* embryonic epidermis.

Among key results, the authors report the detection by live imaging of calcium transients in MCCs prior to their integration into the surface layer of the epidermis. Further, they provide evidence that these transients are generated by PLC activity, and require Calmodulin to be translated into effective actin network construction and apical expansion.

This work is original as calcium signalling has been linked to cell migration in multiple contexts, but not to epithelial integration in vivo. Considering the importance of epithelial integration in health and disease, a much improved version of this study would be of potential interest to a large community of biologists.

We identify several major issues that need to be addressed to complete the study, strengthen the demonstrations, and improve the presentation of the experimental evidence.

1. The authors claim that calcium transients rather than global calcium levels trigger epithelial integration. For this assertion to be strengthened, the characterization of calcium transients needs to be improved. The proportion of MCCs that display calcium spikes should be quantified. Can Gecored positive cells insert without detectable spikes? The number of spikes per MCC over time should be quantified. Do spike number or intensity correlate with the timing of insertion? Do calcium transients also concern other radially intercalating cells, in particular ionocytes that insert soon after MCCs? It is unclear how the shapes of the apical area in F' are defined based on the views in F. We did not find how the normalization of the surface area was done. Fig1G is not convincing with only 3 cells showing significant expansion, considering that apical area expansion is expected to go up with time. We do not understand how surface area can be defined and measured in H'', based on the views in H, since cell borders are not visible.

2. The study is confusing as to whether apical insertion and/or apical expansion depend on calcium signalling. We do not understand the choice of protocol, where drugs are applied at stage 15, several hours before epithelial integration and the functional consequences evaluated at stage 24, while transients are shown to last 20 seconds on recordings done at stage 19. We suggest that calcium transient recording and 2APB treatment should be done at the stage when apical insertion starts, and at the stage when apical expansion is underway. Incidentally, Fig4B,C seems to report PLC pulses in a cell that has already significantly expanded its surface.

3. In relation to the proposed function of calcium transients, it appears important to address whether intercalation and surface expansion is permanently blocked or simply delayed by the various drugs against calcium pathway components. This would inform whether calcium signalling is an absolute requirement for epithelial integration, or if it primarily controls the timing of this process. This question applies to both MCCs and other intercalating cell types, as we can see only goblet cells in Fig2A (2APB treatment).

4. One major problem with the study as it stands is that functional analyses rely only on drugs and dominant-negative constructs. No attempt was made to deplete critical endogenous components of the pathway. A quick survey reveals that Calmodulin (Calm1) is specifically up-regulated in mature *Xenopus* epidermal MCCs (Lee et al., 2023). It would significantly strengthen the study to include Calm1 knockdown to further inform what is the function of calcium signalling in MCCs.

5. The last section on the link between calcium and actin network formation appears as one of the weakest of the study. The potential implication of RhoA is quickly dismissed with a single case and no quantification. The single case phalloidin phenotype in Fig7C-E (again no quantification) is extremely mild compared to those reported in Fig. 6F-N. Fig7G-I does not help to establish a mechanism, but simply confirms that combining two treatments weakening actin filament formation have more effect than each individual one. Here again, the paper would dramatically benefit from the identification of the endogenous link between calcium and actin (Cobl is a good candidate that is up-regulated in mature *Xenopus* MCCs). This would possibly offer the possibility to overcome defects induced by calcium drugs by overexpression of the most downstream effector of the pathway.

Minor issues:

- Movies must be annotated to help the reader understand the contents. Time should be displayed. We could not identify captions for the movies.
- Movie 1 is bugged with a reiteration at about 17s
- Many typos are present throughout all sections of the manuscript and in figure legends.
- In many places, more care should be given to the wording; e.g. legend 2H mentions the various steps of epithelial insertion, while Fig2H describes epithelial insertion as one of the discrete steps.
- The mat/Met section is very minimalist, lacking for instance all commercial references for the various drugs used in the study.
- In quite a few places, the authors make logical leaps that are problematic. Fig3D does not establish that PACR-positive cells are MCCs. Fig5F does not establish that CALM-positive cells are MCCs, particularly those on the left column (bottom arrow must be repositioned). In Fig7A the authors use rGBD staining as a proxy for basal bodies, which is not acceptable.

Version 1:

Reviewer comments:

Reviewer #1

(Remarks to the Author)

The authors have added a number of experiments that have improved the manuscript.

I remain concerned that much of their data is entirely dependent on the 2-APB reagent. Their results do suggest that Calcium plays a critical role; however, whether it is calcium transients is entirely dependent on the 2-APB being specific to calcium transients and not basal calcium levels. They do add data suggesting that basal calcium levels appear unaffected.

Therefore I am inclined to accept their results and see if others can build on their results and augment the evidence.

Reviewer #2

(Remarks to the Author)

This is a much improved paper and I am generally supportive. However there are still some unresolved issues.

1. "Intensity profiles are used widely in the literature...." They are used widely when done properly. The paper has now quantified some of the line scan intensity profiles which is an improvement. However, the "data" in figure 5A-B and 6K are still meaningless. In 5A-B they draw a line and show the profiles of the fluorescent intensity as if this is important, but one could draw a different line in the same cell and get a totally different profile that does NOT show actin/calm between two centrioles. This is minor point but I think 5B should be removed. 6K is the same situation, an n of 1 is not an experiment. You know how to quantify fluorescent intensity as you have done in many of the other figures. If you want to include the tubPACR data then do it right or remove it. S9C meaningful, S9B not meaningful.

2. In my initial assessment I commented on how sloppy Figure 6D was. I thought it was obvious but clearly I did not explain

my concern. Please see the images attached which the authors claim are the same image but with Utr (left) and Geocode (right). Clearly they are not aligned properly. See the opacity overlay to see how out of sync they are.

See attached figure.

3. My concern regarding figure 4H (now 4G) remains. I originally tried to find the definition of apical insertion because of this image. Now that you have defined apical insertion as 10um², I am even more concerned. For the image on the right, if your scale bar is accurate, the MCCs is going to be over 10um² in my estimation. Its 25um long and at least 1 wide. I am not saying that this cell has inserted properly by any means but this representative image calls into question the methodology.

Reviewer #3

(Remarks to the Author)

I would like to commend the authors for the extensive revision work with several new decisive experiments that help strengthen their conclusions. I find particularly important the new demonstration that 2APB also inhibits apical emergence of other basally localized cells (ISCs and SSCs), which indeed suggests that calcium transients may be more generally involved in epithelial integration in development and homeostasis.

All of my issues have been satisfactorily addressed. I do understand the challenge represented by my request to identify endogenous players by genetic knockdown, and do not consider this a blocking point, considering the numerous improvements made to answer the questions raised by all three reviewers.

The paper has been greatly improved and now represents an original and convincing piece of work that should be of interest to many cell and developmental biologists.

Reviewer#1

The manuscript by Chistodoulou and Skourides addresses the mechanism by which basal cells insert into an epithelial sheet. In *Xenopus*, the embryonic epidermis acts as a mucociliary epithelium in which multiciliated cells (MCCs) originate at the basal layer and then must migrate apically and insert into the overlying epithelium. There are many interesting processes here that are not well understood. For example, once the apical tip of the MCC hits the epithelial surface, it must create pushing forces that expand laterally to insert the cell into the epithelium. Additionally, during this process, apical actin creates a “honeycomb” network in which basal bodies dock establishing proper spacing between cilia and defining the correct density of cilia/surface area. Therefore, these MCCs are extremely interesting from their initial specification, apical intercalation, and formation of planar-coordinated beating cilia in an epithelial sheet.

The authors seek to better understand the role of calcium transients in the intercalation of the MCC into the epithelium. The authors identify a phospholipase C-calcium transient – calmodulin pathway that is essential for actin stabilization. They argue that prior to apical emergence of the MCC, calcium transient(s) is/are essential for the stabilization of actin via calmodulin, that can create the pushing forces for lateral expansion of the apical MCC surface.

We would like to thank the Reviewer for the thoughtful and constructive feedback. The Reviewer’s comments have helped us to strengthen our initial findings.

MAJOR CONCERNS:

1) Calcium transient vs. steady-state calcium concentrations

The authors begin by using a RGECO calcium sensor and detect flashes of fluorescence in MCCs. The main point of Figure 1 is that Ca transients occur before intercalation as measured by apical surface area. I have a number of concerns with this conclusion:

a) the data presented in Fig 1G suggests that the apical surface area increases mildly after a calcium transient. There are three outliers but really the increase is small in the majority of the data before or after calcium transient with most changes from 1.2-1.5 fold with significant overlap before and after the transient (of note the apical area of an intercalating MCC increases dramatically – many fold increase

This is a valid point stemming from the restricted window used for live imaging. Specifically, we quantified the surface area increase over a time period of 1 hour (30min before a calcium transient and 30 minutes after a calcium transient) in the same cell. However, with this approach, we might be missing calcium transients that were generated just before the 30min time window before the appearance of a calcium transient. Thus, in the revised version of the manuscript, we provide additional quantifications, assessing the impact of calcium transients on apical area expansion. Specifically, we compared the surface area variation in cells that displayed a calcium transient

to those that did not during time-lapse recordings. In this analysis, it becomes evident that the surface area increases drastically when a cell displays a calcium transient. These data are presented in Figure 1I of the revised manuscript (see image on the right).

I also note that plots of the calcium transient vs the surface area of the MCC don't correlate very well. Looking at supplemental figure 1B – panel 1 looks like the increase in surface area occurs 10 min after the transient, in panel 2 concurrent with the expansion, panel 3 10 min after, panel 5 seems to have no change in the acceleration of apical area with two transients. All of these panels have different time windows, and dramatically different fold increases (panel 1, IV 2 fold, while panel 3 is 10 fold). The lack of consistency makes it difficult to correlate the flash with the amount of change in apical area that is essential for their argument of correlation.

We appreciate the Reviewer's observation regarding the variability of the plots. The plots of calcium transients vs. surface area are not expected to be identical for each cell. Each apical emergence event is cell autonomous, as shown in various studies. Individual MCC will be exposed to a distinct mechanical environment based on their insertion site. This will affect the timing and extent of apical surface area expansion following a calcium transient.

To address this issue, we provide a new analysis and revised Supplementary Figure 1. In the previous version of the manuscript in Suppl. Figure 1 we provided data showing the surface area over time for MCCs at different steps of apical emergence. We understand that this can lead to confusion. We have revised the Suppl. Figure 1 and presents plots of MCCs during the initial stage of apical emergence, during which the surface area increases dramatically because cells acquire apical surface area and expand it. In addition, we present representative plots of cells during apical surface area expansion, with 1, 2, 3 and 4 calcium transients (Supplementary Figure 1 below).

Moreover, quantification of surface area increase in cells that display and do not display a calcium transient shows a clear correlation between calcium transients and successful apical emergence (Figure 1i revised manuscript, see previous response).

Additionally, a major concern is the time resolution. It appears that while the movies are taken for long periods of time (30 min) the resolution is low – every 30-40 seconds

for transients whose period is not clear. Are there short flashes that occur for 15 seconds (still a substantial flash) that might be missed? How long are these flashes?

We have quantified the duration of calcium transients in the first version of the manuscript from time lapses with a 20-second time interval. Quantification is shown in Figure 1D of the revised manuscript.

From time-lapse recordings of 15-second time intervals, we could not detect more transients of shorter lifetimes. In our time-lapse recordings, we perform live imaging of a live 3D tissue. In order to image superficial and deep cells in each movie to capture apically emerging cells, we use a large field of view, and we generate z-stacks with a large number of optical sections due to embryo curvature. Therefore, we cannot perform time-lapse recordings of higher time resolution with our current infrastructure.

How many flashes occur per cell?

We have now quantified the number of calcium transients per MCC, showing that the majority of MCCs display more than one calcium transient during apical emergence. These data are presented in Figure 1E of the revised manuscript (see below).

I worry that they may be missing some that might indicate that these flashes are not exclusively occurring prior to the intercalation which could change their interpretation as a signal for epithelial insertion or as pulses that drive waves of actin formation. For example, in the context of PLC/IP3 inhibition, perhaps the role of these molecules is to intensify a flash that is coming from another source. Inhibition with 2APB might still lead to flashes, but these become too short (to detect in their 30 s sampling) and weaker without PLC/IP3 intensification.

Our time-lapse imaging was performed at stages before MCCs acquire an apical surface and during MCCs apical emergence. Thus, as we show in Figure 1I-J, the presence of these flashes is correlated with MCC apical emergence and surface area expansion.

The quantification of calcium transients in 2APB and PLC inhibitor-treated embryos shows that calcium transients' generation is defective (Supplementary Figure 1A., Figure 4D, see below).

We never observed transients of lower intensity in embryos treated with PLC inhibitor and 2APB. Therefore, we have no reason to believe that the duration of calcium transients is shorter upon inhibition of IP3R and PLC.

b) To test the importance of these transients, the authors seek to inhibit them with 2APB or PACR. While the authors quote a single study stating that 2APB alters calcium transients but not basal intracellular calcium level, they do not measure that in this context. The previous study examined the neural plate and neural plate closure, a very different process than apical intercalation of MCCs. They simply cannot make the argument that MCC intercalation is dependent on calcium transients without confirming that basal calcium levels are unaffected (as was done in the quoted study). We know that actin polymerization depends on calcium and so depletion of basal calcium levels will impair actin which could entirely explain their results. To make the claim that calcium transients drive apical intercalation and not basal calcium levels, the burden of proof is on the authors to demonstrate no change in basal calcium levels.

We thank the Reviewers for this suggestion. We have performed experiments to assess the impact of 2APB treatment on basal calcium levels. We believe that these new data significantly strengthen our initial conclusions. Specifically, to examine if basal calcium levels are affected by 2APB treatment, we used the ultrasensitive ratiometric Ca^{2+} indicator yellow Cameleon-Nano (YC-Nano) (Horikawa et al., 2010). Using spectral imaging, we quantified the FRET/CFP ratio upon excitation with a 458 laser (CFP excitation) in MCCs from control and 2APB treated embryos. This revealed that FRET efficiency, which represents the intracellular basal calcium levels, is unaffected by 2APB treatment, in agreement with previous work in *Xenopus*. These data are presented in Supplementary Figure 2B-D of the revised manuscript (see below).

c) Measuring calcium transients vs basal calcium levels is not a trivial task and requires the use of different calcium sensitive dyes that can detect the different levels with sufficient signal-to-noise. It is also critical that this be done ratiometrically as the literature is fraught with complications where transient calcium imaging leads to artifactual results because the “load” of calcium dye is not controlled for from cell to cell. This is a major weakness of this paper.

In the revised version of the manuscript we provide data generated using the ultrasensitive ratiometric Ca^{2+} indicator yellow Cameleon-Nano in combination with spectral imaging to quantify basal calcium levels ratiometrically. See the previous comment and response.

d) no data is presented that 2APB actually eliminates calcium transients. There is simply data showing that the MCCs are not intercalated (an end result). There still could be calcium transients with the 2APB treatment. I agree there is a dramatic effect on MCCs with 2APB treatment but the authors need to confirm that this leads to a loss of calcium transients (and normal basal calcium). Otherwise they simply cannot make a conclusion.

In our initial submission, we provided data showing the occurrence of calcium transients in control and 2APB treated embryos. These data are presented in Supplementary Figure 2A of the revised manuscript. Additionally, in the revised version of the manuscript, we provide a time-lapse recording from a representative 2APB treated embryo expressing GECO-RED (Movie 4).

e) PACR data does not resolve this challenge as PACR could very well chelate both basal and transient calcium concentrations.

We agree with the reviewer that PACR will chelate calcium within a cell; however, PACR was utilized here as a tool to demonstrate that intracellular calcium is necessary for MCC epithelial insertion in a cell-autonomous manner.

In the revised version of the manuscript, we show that the defects observed upon 2APB treatment cannot be attributed to changes in basal calcium levels, as these remain unaffected in 2apb-treated embryos (Supplementary Figure 2B-D).

I note the authors do not take advantage of the optical properties of PACR. If the embryos are bathed in blue light, does the MCC phenotype rescue?

We thank the Reviewer for suggesting this experiment. According to the Reviewer's suggestion in the revised version of the manuscript, we provide data showing that exposure of embryos expressing PACR to blue light (460 nm), rescues the PACR phenotype. These data are presented in Figure 3E and Supplementary Figure 5 of the revised manuscript (see below, Figure 3E left, Suppl. Figure5 right).

Can they use the PACR photosensitivity to titrate the chelation to inhibit transient calcium, but not basal calcium?

While this experiment could be exciting, we cannot see how it can be experimentally feasible. During a calcium transient, calcium levels are higher than basal calcium levels. Thus, titrating PACR photosensitivity to only inhibit calcium transients is not possible.

f) another option would be to load the cells with caged calcium, inhibit with APB, and then release the calcium with a pulse of light to simulate a transient.

The reviewer is correct that if this was technically possible, it would be informative. Unfortunately, using caged calcium to simulate transient calcium transients during MCC apical emergence is not feasible. Caged calcium loading will lead to the loading of caged calcium in all cell types, including goblet cells. During apical emergence, MCCs are found in the deep epithelial layer without a distinct apical domain. Uncaging calcium during apical emergence will lead to calcium uncaging in goblet cells. The lack of spatial control for calcium uncaging will not allow us to restrict calcium level increase in MCCs. Thus, it would be impossible to emulate the physiological calcium transients essential for proper apical emergence.

This would be better than thapsigargin, which is stated to be in figure 6d-e but is completely unlabeled in the experiment or lacks control.

We don't understand why the Reviewer claims that Figure 6D is unlabeled. We ensure that in the revised submission all figure panels are labeled. The relevant time-lapse recording showing concomitant calcium increase and apical actin enrichment is provided in Movie 10 of the revised manuscript.

As a control for this experiment, we present data showing apical actin levels over the same time period in MCCs from control embryos in the revised manuscript (Figure 6E, below). These data demonstrate that actin enrichment in a short period is only induced in the presence of thapsigargin as a result of elevated calcium levels.

2) Finally, the source of the calcium transients is not clear. The authors suggest that these transients are coming from the ER which is certainly possible but they also could be triggered across the plasma membrane and then amplified by PLC/IP3/ER. There are both voltage gated channels and mechanosensitive channels that could be playing a role at the membrane surface. The source of calcium transients, based on our data, is PLC activity (Figure 4A-D, Movie 7-8). Mechanosensitive gated calcium channels do not affect MCC apical emergence (Kulkarni et al 2021), and therefore these channels cannot be responsible for calcium transient generation during MCC apical emergence. In addition, voltage-gated calcium channels are not expressed in MCCs during mucociliary epithelium development (image below generated from https://kleintools.hms.harvard.edu/tools/currentDatasetsList_xenopus_v2.html, Briggs et al., 2018)).

In agreement with the latter inhibition of voltage-gated calcium channels, with the voltage-gated channel inhibitor Mibefradil (Viana et al., 1997) does not cause defects in MCC apical emergence. Specifically, we treated stage 15 embryos with 20µm Mibefradil, and subsequently we examined MCC apical emergence in stage 25 embryos. MCC apical emergence was not affected by Mibefradil treatment (see image below).

Overall, based on these data, I am unconvinced that the authors can claim that calcium transients are critical for MCC intercalation. I don't doubt that calcium is necessary for actin to form the apical actin network and that inhibiting IP3/PLC will reduce the actin, lateral pushing force, and MCC intercalation.

We believe that since we present data that 2APB treatment only affects calcium transients' generation and not basal calcium levels, we can claim that calcium transients are critical for MCC apical emergence.

However, whether this is due to a transient calcium flash is unclear. Logically, what is the purpose of the 30 sec pulse of calcium? The intercalation of the MCC occurs over 1-2 hours and so what would be the purpose of a 30 second calcium transient – certainly this is not a sustained calcium source necessary for the continued polymerization of actin which must be occurring with basal calcium levels over this time period.

Transient increase of calcium, and calmodulin activation can lead to a cascade of events leading to downstream effector activation either by conformational changes upon direct interaction with Calmodulin or by post-translational modifications such as phosphorylation. The lifetime of these changes in downstream effectors can be in the timescale of hours, translating a short-term event into a long-term process.

Our data also show that during apical emergence, MCCs display more than one calcium transient, showing that this cascade can be enhanced during apical emergence.

Therefore, translation of short-lived calcium increases to cellular behaviours of higher time scales can regulate apical emergence. We include this hypothesis in the discussion section of the revised manuscript (lines 34-45, page 10).

However, if that is the case, what is the mechanism? What actin related proteins are “snapped” due to this calcium transient that lead to sustained polymerization (memory of the pulse) over 1-2 hours? This certainly would be very novel and of interest to a broad audience like Nature Communications.

We agree with the reviewer that this is an interesting question, but it is beyond the scope of this study. Identifying possible changes in the actin regulatory proteome composition in response to calcium elevation during apical emergence is a challenging, long-term task. To answer this question effectively would require years of work to identify the actin-binding proteins that regulate actin polymerization subsequent a transient calcium increase. This is a project on its own and cannot be a small part of the revisions for the current manuscript, in which we identify for the first time the role of PLC/Ca²⁺/Calmodulin signalling axis in apical emergence.

But they must prove that basal calcium levels are unaffected, define the sufficiency of the calcium pulse, and the requirement of the calcium pulse over basal levels.

We now present data showing that calcium transients and not basal calcium levels are necessary for MCC apical emergence (Supplementary Figure 2). See relevant response above.

Minor Issues

Fig 1 – In fig 1B- the authors label four cells that they then show a kymogram in Fig 1C. It would be very helpful if the labeling in the top right panel of the Fig 1B indicating the four cells were reproduced in all the other panels at different time points. While I think I can decipher which cell is which, it would be very helpful to have each panel labeled with the four cells.

We label the 4 cells at all different time points in the revised Figure 1.

Supplementary figures and movies – please label each in the figure or in the movie whether they are Supp Fig 1, 2, 3, etc or Movie 1, 2, 3. Movies are labeled as 556027_0_video_9850791_smf468- and so I just guessed that 791 was Movie 1 and 792 was movie 2, etc. For the supp figures, again I just guessed on order – the main figures are nicely labeled with Figure 1, 2, 3, etc in the margin - please do the same with supp. Figures and movies.

In the revised manuscript, we label each Suppl. Figure and Movie as suggested by the Reviewer.

Reviewer #2 (Remarks to the Author):

The manuscript by Christodoulou et al. addresses the important question of the role of calcium signaling during radial intercalation. The authors identify calcium transients that precede the ability of cells to apically insert into the epithelium. They go on to use both pharmacological and dominant negative approaches to test the function of intracellular calcium release (2APB and PACR), PLC (U73122 and PLCD1), and Calmodulin (W7 and CALM1234). Finally, they assess the

consequence on apical actin during 2APB treatment. Overall they propose that Ca^{++} \diamond PLA \diamond CALM axis regulates both apical insertion and apical expansion. This paper addresses complex questions and the overall message that Ca^{++} signaling is important to the process is believable. However, much of the data is of poor quality, the presentation is sloppy and there is a significant amount of over interpretation that drives my enthusiasm significantly down. This is a mature field and the authors have notably attempted to place their work in the broader context but in doing so have created a poorly focused message that is difficult to interpret. That said there are also some interesting findings and with significant work this could be an impactful addition to the field.

One of my primary concerns is that they claim that Ca^{++} is important for both insertion and expansion. However, the timing of the experiments make this claim difficult to determine as this is an ongoing process and a delay in insertion will appear as a loss of expansion when really the process is just delayed. To make these claims the experiments would either need to be done more dynamically or at the very least be done past the normal end of the process to capture any delays. As presented, I don't think these separate claims are warranted, but the claim that Ca^{++} is involved in the overall process is justified.

We thank the reviewer for the constructive criticism and for acknowledging that our work can be an impactful addition to the field. We believe that the revised version of the manuscript fully addresses the reviewer's comments. Specifically, we performed new experiments of dynamic 2APB treatment to address the reviewer's concern on the necessity of calcium transients for both MCC epithelial insertion and surface area expansion. In addition, we provide new data showing that the observed defects are not due to delays in the process. Furthermore, we provide new data and data analysis in combination with text and Figure revisions to fully address Reviewers' comments.

Comments:

Mandatory. The process being studied has been repeatedly termed radial intercalation which has also been called apical emergence being broken down into two steps apical insertion and apical expansion. For the continuity of the field you need to refer to this process as radial intercalation or apical emergence rather than epithelial integration in the title, abstract and throughout the paper. While I don't mean to curb your creativity, rebranding of processes is a scourge on science making it harder for researchers to follow the field.

We didn't have any intention of rebranding an already known process. The term of epithelial integration has been used in a recent publication (Ventura et al., 2022) and we felt that this term better describes the whole process compared to the term radial intercalation.

This is because the term radial intercalation cannot differentiate between the insertion of cells into a fully polarized epithelium with tight junctions (MCCs during mucociliary skin epithelium development) and the insertion of cells into an epithelial layer with no tight junctions (radial intercalation of ectoderm cells during epiboly).

For consistency with previous studies, we now use the term apical emergence in the revised manuscript according to reviewer's suggestion which we believe better describes all the steps of the studied process.

From the analysis (e.g. 1C, D) it would seem as if there is a single Ca⁺⁺ spike, but the movies make it look like individual cells have multiple spikes. Can you comment on this and whether it takes repeated burst to induce intercalation.

The analysis in Figure 1C and D of the initial submission utilized a narrow time window to specifically highlight the occurrence of individual calcium transients in MCCs. Indeed, the majority of MCCs present multiple calcium transients during apical emergence. We quantified the number of calcium transients per MCC during apical emergence. These data are presented in Figure 1E of the revised manuscript (see below).

Figure 1G “Quantification of MCC surface area increases in a 30min time window before and after the appearance of calcium transient”. Given the stage of this analysis and the length of the measurement one would expect cell size to change so it is hard to assign function to the Ca⁺⁺ transient. Could you compare the size change in cells that do or do not have a Ca⁺⁺ transient?

We thank the reviewer for this suggestion. We have quantified the size change in cells with and without a calcium transient, and we show that apical surface increase is evident only in cells displaying calcium transients. These data are presented in Figure 1I of the revised manuscript (see Figure below). We believe that these data better illustrate the correlation of calcium transients with MCC apical emergence.

F' Shows the cell size at two points from the movies presented in F, but it is not clear how those measurements were made as the cells do not look considerably different in the images. This calls into question the data in F'. At the very least show an outline of what is being measured. Better cell boundary imaging would help.

Imaging 3D morphogenetic events with high temporal resolution is difficult, and sometimes, to achieve the high temporal resolution necessary for imaging calcium transients, you must compromise spatial resolution. We now show the cell boundaries in Figure 1G of the revised manuscript to help the readers understand how surface area measurements were made.

Statistics in 1G should be a paired t test and overall not enough statistical information is provided throughout the manuscript.

The data are now presented in Figure 1H. The statistics used for Figure 1H was a paired t-test. In the previous version of the manuscript, we had a typing mistake in the Figure legend. This has now been corrected. We now provide statistical information throughout the manuscript in all relevant Figure legends

The imaging in 1H is so low quality as to be uninformative.

These data are now presented in Figure 1J. We provide close-up images in the revised manuscript to make the Figure more informative (see below).

2APB is not a specific drug making interpretations of its use difficult. Particularly as it also affects TRP channels which are likely involved in this process. I think other experiments validate this finding but this should at least be stated and discussed rather than calling it a specific blocker.

This has been discussed in the text of the revised manuscript. Lines 34-35 page 4, “we employed 2-aminoethoxydiphenyl borate (2APB), which blocks IP₃R and other membrane-localized Ca²⁺ channels”.

Figure S1C does not have a scale bar and it is hard to see what this is. If it is super low resolution as I suspect then it is not clear what is being quantified. Also the cells in D are not identified. Also quantification doesn't provide n's.

We now provide the time-lapse recording instead of images to better show the effect of 2APB treatment on calcium transients generation (Movie 4). The number of cells used to quantify the effect of 2APB in calcium transients is now provided in the figure legend of Supplementary Figure 2A.

The claims in Figure S2 are problematic. First it is mislabeled making it confusing, but if I understood things correctly I would say that for bCatenin, the staining looks quite different in control versus 2APB, however there are quite a few cells that have intercalated so I am not sure that the drug treatment even worked. Regardless single line scans is not sufficient to make any claims.

These data are now presented in Supplementary Figure 4. The revised Figure is correctly labelled. Histograms of fluorescent intensity are used in many studies to highlight the localization of proteins at specific cellular compartments. However, to address the Reviewer's concern, we have now quantified the signal of markers used in Supplementary Figure 4 along a line passing from the cytoplasm to the cell cortex. The quantification was done from 30 different cells of 3 different embryos. Our quantification and multiple t-test statistical analysis shows clearly that cortical enrichment of E-cadherin, beta-cat and ZO-1 is unaffected by 2APB treatment (see Supplementary Figure 4 below).

Regarding the Reviewer's concern that in this experiment, 2APB treatment didn't work, we want to state that in the images provided for b-catenin, they are 8 MCCs apically emerged in the field of view from an embryo treated with 2APB (these MCCs have a very small surface area). In control embryos in the same field of view, there are 20 MCCs with fully expanded surface areas. This clearly shows that 2APB treatment worked in this experiment.

Lines scans of relative intensity for a single cell are simply not informative. This type of data is repeatedly used throughout the paper.

Intensity profiles are used widely in the literature. They provide a quantitative way to analyze the spatial distribution and relative abundance of the labeled target molecules along the line or across the region of interest. The profile can show where the labeled target is concentrated within a cell or tissue. Peaks in the intensity profile correspond to areas with high concentrations of the target molecule, while valleys indicate areas with low or no signal. When multiple targets are labeled with different fluorophores, you can generate intensity profiles along the same line for each channel. Overlapping peaks in the profiles suggest that the different targets are co-localized in the same cellular or subcellular regions. The height of the peaks in the intensity profile can provide a semi-quantitative measure of the relative abundance of the target molecule in different compartments or cells within the image. Higher peaks generally indicate a higher concentration of the labeled target. In addition, the slope of the intensity change at the edges of a structure can indicate the sharpness of its boundaries. A steep slope suggests a well-defined boundary, while a gradual slope might indicate a more diffuse distribution. Finally, by analyzing intensity profiles across multiple cells or regions, you can assess the heterogeneity of target expression or localization within a population. Specifically, we used Intensity profiles to highlight Calmodulin's localization in MCCs (Figure 5B), to highlight the loss of apical actin enrichment in PACR+ MCCs (Figure 6K), to highlight the loss of apical actin enrichment in Suppl. Figure 9B and to assess the effect of 2APB treatment on adherens junctions and tight junctions components localization (Suppl. Figure 4).

The description of Figure 2C does not really add any new information to the field and should probably be supplemental, but it should absolutely reference the work that showed these steps first, in particular Sedzinski 2016 and Ventura 2022.

We didn't have any intention to argue that these are new findings by our research. We used this panel as a reference so that a reader could familiarize themselves more easily with the different steps of the process. Sedzinski 2016 and Ventura 2022 were cited in the first version of the manuscript. We have now moved this data in Suppl Figure 3D, and work relevant to the different steps of the process is also cited in the specific paragraph (lines 4-10 page 5) of the revised manuscript.

The authors claims that "our data indicate that intracellular calcium transients are indispensable for MCC epithelial integration and apical cell surface expansion (Figure 2H)." But since the 2APB blocks insertion it is hard to say what affect it is having on expansion. Cells that don't insert cannot expand. But also as mentioned above these effects could be delays.

We thank the reviewer for this comment. We think that the new experiments presented in the revised version of the manuscript clearly show that calcium transients are necessary for both MCC epithelial insertion and apical surface area expansion. In addition, we present new data showing that these defects cannot be attributed to a delay in the process. Specifically, we treated embryos with 2APB from stage 15 to stage 28 (well beyond the completion of MCC apical emergence) and MCC apical emergence is defective. This experiment indicates that the defects observed upon inhibition of calcium transients are not due to a delay in the process, which is completed at stage 24. These data are now presented in Suppl. Figure 3A-C of the revised manuscript (see below).

Furthermore, to examine the role of calcium transients during apical cell surface expansion, we treated embryos with 2APB at the beginning of apical emergence (stage 19) and after most of MCCs have inserted and undergo apical cell surface area expansion (stage 22). These experiments revealed that even though late treatment with 2APB does not affect the insertion of MCCs into the superficial epithelium, it severely affects the expansion of MCC's apical surface area. These data are presented in Figure 2 F-G and Suppl. Figure 3 G-I of the revised manuscript (see below). In conclusion, these data show that calcium transients are necessary for both insertion into the epithelium and the subsequent surface area expansion.

It is not clear why the authors used the Ca⁺⁺ caged optogenetic PACR, as they do no optogenetic experiments. If the cells could intercalate after blue light treatment that would be informative.

Initially, we used PACR as a genetically encoded calcium chelator. In the revised version of the manuscript, we present data showing that exposure of embryos expressing PACR to blue light (460 nm) rescues the cell autonomous apical emergence phenotype induced by PACR overexpression. These data are presented in Figure and Suppl. Figure 5 (see below).

Figure 4B is hard to interpret as presented. What step of the process is the cell at, as it looks as if it is already intercalated? The overall cell shape and what looks like the apical surface does not change over the 40 minute movie? It is not clear what they are proposing is happening here or what the consequences are.

We performed new time-lapse experiments of stage 19 embryos expressing the PLC biosensor. These new data clearly show pulsed PLC activity during MCC epithelial insertion and apical surface area expansion. These data are presented in Figure 4A-B(see below) and Movie7 of the revised manuscript.

U73122 has been reported to have numerous off target effects. While this result is largely validated with the PLCD1 it should still be stated.

We now revised the text to include a statement about the potential off-target effects of U73122. Lines 40-41 page 6.

Does Thapsigargin promote early apical insertion as would be predicted by your model? This would really strengthen your claims.

While this experiment would indeed support our claims, Thapsigargin treatment affects calcium levels in all cells of the epidermis. Therefore, we cannot perform long-term experiments using Thapsigargin to assess the impact of calcium level increase in the timing of MCC apical emergence.

“Inhibition of PLC suppresses calcium transients (green circles) generation.” The overall GECO-RED signal appears much higher in the U73122 treated tissue which would confound this interpretation. Please explain.

We apologize for this confusion. The signal of GECO-RED in PLC-treated image was stretched to highlight the expression of GECO-RED. We have now revised the figure without stretching the signal of GECO-RED in PLC-treated images (Figure 4C, see below). In addition, we now provide the representative time lapse recording (Movie 8) to help the readers assess the effect of PLC inhibition on calcium transients' generation, and we provide quantification for the number of MCCs displaying calcium transients in control and PLC-inhibitor treated embryos (Figure 4D, see below).

N's should list embryos tested not just cells. This is true throughout the paper and analyzing 20 MCCs from the same embryos is not the same as analyzing 5 MCCs from 4 embryos.

We now refer to the number of embryos tested for each experiment in the relevant figure legends.

Many of the figures are sloppy. Perhaps the worst example of this is that the images in 6D are not properly matched with Utr and GECORED.

The labels in 6D were matching the Utr-GFP and GECORED signals in our initial submission. We are not sure why the reviewer thinks that the labels don't match the signals. In the revised version of the manuscript, we have revised all the figures to present the data in a better way.

The cropped images only partially match.

We don't know to which cropped images the Reviewer is referring to. In the revised manuscript, we provide revised Figures for all main and supplementary Figures.

"Thus, our data suggest that the defects in MCCs epithelial integration upon inhibition of calcium transients do not stem defective centriole amplification." I am not sure these data totally rule out this possibility but a bigger concern is that it is known that centriole docking facilitates the establishment of the apical actin (Kulkarni et al 2018) and the data in S4B shows that the basal bodies are not docked, which would alter actin.

The data from Figure S4B of the initial submission are now presented in Supplementary Figure 8B. These data show that basal bodies are not docked because these MCCs do not acquire an apical surface upon 2APB treatment. With these images, we wanted to show that the apical translocation of basal bodies, which depends on apicobasal polarity, is not affected. Therefore, in agreement with our data regarding PAR3 localization, these data show that the apicobasal cell polarity of MCCs is not affected by 2APB treatment.

When we use a lower concentration of 2APB, MCCs insert into the superficial epithelium while MCC surface area expansion is affected (Figure 6G) and the basal bodies are docked (Figure 7A-D). Therefore, apical emergence defects in the absence of calcium transients cannot be attributed to defective basal body docking.

In agreement with our claims, published work (Kulkarni et al, 2021) showed that pharmacological disruption of centriole amplification does not affect apical emergence but only affects the regulation of apical surface area size, with MCCs devoid of centrioles presenting larger surface area compared to control counterparts. In addition, defective basal bodies docking only affects ciliogenesis but not apical emergence (ELMO1 and Rac1, Epting et al., 2015; Dishevelled, Park et al., 2008; Foxj1, Stubbs et al., 2008). All the above show that while basal body docking and apical emergence occur at the same developmental stages, the two processes are uncoupled.

There is a complete lack of continuity on how experiments were quantified (or not). Data such as 2G should be quantified.

Data from 2G are now presented in Suppl. Figure 3E and are quantified (Supplementary Figure 3E-F).

Nowhere could I find a description of how insertion was defined for the quantifications of inserted or not.

We have now added a description of how successful insertion was defined in the methods section. 'Successful epithelial insertion was considered when cells displayed an apical surface area $> 10 \mu\text{m}^2$ '

For example does 4H show two cells that fail to insert or one that has inserted (ii).

These data are now presented in Figure 4G. These data show two MCCs that failed to insert into the superficial epithelium since their apical surface area is smaller than $10\mu\text{m}^2$.

Others have shown that ac tub marks cells prior to intercalation (Collins 2021), but in 3G there is no acetylated tub in PACR cell. If there is really no acetylated tubulin then the cells would fail to insert based on that.

These data are presented in Figure 3D of the revised manuscript. Because the signal of Ac-tub in the ciliary axoneme is very high, sometimes it is very difficult to see the acetylated tubulin in cells that failed to apically emerge without overexposing the acetylated tubulin signal in the ciliary axoneme. The data presented in the revised manuscript (Figure 3D, see below) reveal the presence of Ac-tub in cells expressing PACR. The defects in MCCs upon PACR expression were also validated by atub: PACR, which is expressed only in MCCs after their specification.

Figure 5D does not state stage which is essential for interpretation.

Embryos examined at 5D were stage 32. The stage is now included in the relevant Figure legend.

It is not clear why the cartoon in 5C has CALM asymmetric to centrin when the example in B looks fairly symmetric.

In the initial version of the manuscript, we thought that not overlying the CALM and centrin in the schematic would help someone to see the localization of CALM at

centrioles. However, we understand that this might lead to confusion. We have now changed the schematic showing that CALM and centrin colocalize.

Figure 6A. I am unclear of the interpretation as Utrophin has been shown to be enriched apically (Yasunaga et al 2022). Why is its enrichment transient and why is it not enriched prior to the Ca⁺⁺ pulse. The way this data is presented is confusing.

We have changed the presentation of the data to avoid confusion. Indeed F-actin is enriched apically, as shown in the revised Figure 6A-B (see below). This enrichment is enhanced transiently upon the appearance of a calcium transient and is followed by apical surface area expansion (Revised Figure 6C, see revised Figure panels below). Thus, our data show that calcium transient appearance is temporally correlated with changes in the apical actin network during MCC apical emergence.

What Stage in 6D. Are these cells inserted already? Not sure what the relevance is here to the overall claims of the paper.

We now state the stage of the embryo in the Figure legend. We performed this experiment in stage 21 embryos because we wanted to examine the impact of calcium increase on the actin network. For this experiment, to be done it was essential for MCCs to first acquire a small apical area.

Minor:

For figure 1 the text refers to stage 18 but the figure states stage 19.

This has been rectified in the revised manuscript.

Why was neuroepithelium used in 4F and what exactly does that mean.

This was a mistake. This has been corrected in the revised manuscript.

Figure 1B should continue to label the 4 cells that are being tracked.

We now show the 4 label cells in all frames.

No scale bar references in the legends.

In the revised version of the manuscript, we provide scale bars and scale bar references for all Figures.

Centrosomes are comprised of two centrioles which are identical structures to basal

bodies (not similar to as the text states). Figure 5C should be labeled centriole not centrosomes.

We are sorry for this typing mistake. We changed the term centrosomes to basal bodies.

Reviewer #3 (Remarks to the Author):

In this paper, Christodoulou and Skourides investigate the role of calcium signalling in radial intercalation of multiciliated cells (MCCs) in the *Xenopus* embryonic epidermis.

Among key results, the authors report the detection by live imaging of calcium transients in MCCs prior to their integration into the surface layer of the epidermis. Further, they provide evidence that these transients are generated by PLC activity, and require Calmodulin to be translated into effective actin network construction and apical expansion.

This work is original as calcium signalling has been linked to cell migration in multiple contexts, but not to epithelial integration *in vivo*. Considering the importance of epithelial integration in health and disease, a much improved version of this study would be of potential interest to a large community of biologists.

We identify several major issues that need to be addressed to complete the study, strengthen the demonstrations, and improve the presentation of the experimental evidence.

We thank the Reviewer for recognizing that our work will be of interest to a large community of biologists. The Reviewer's comments helped us improve our initial findings and provided us the opportunity to assess the role of calcium in other cell types undergoing apical emergence during the development of the *Xenopus* skin epithelium. We believe we have addressed all the Reviewer's comments through new experiments, data analysis, text and figure revisions, and detailed responses to the Reviewer's feedback.

1. The authors claim that calcium transients rather than global calcium levels trigger epithelial integration. For this assertion to be strengthened, the characterization of calcium transients needs to be improved. The proportion of MCCs that display calcium spikes should be quantified. Can Gecored positive cells insert without detectable spikes?

The proportion of MCCs displaying calcium transients was detailed in the previous version of the manuscript. These data are shown in Suppl. Figure 2A of the revised manuscript

We do not observe cells emerging apically without the appearance of a calcium transient. An example is presented in Figure 1J.

Furthermore, in the revised version of the manuscript, we provide data showing that basal calcium levels are not affected by 2APB treatment (Suppl. Figure 2B-D) in

contrast to the generation of calcium transients, which is severely affected (Suppl. Figure 2A, see below).

Altogether, these data indicate that the defects in MCC apical emergence observed in embryos treated with 2APB stem from the loss of calcium transients.

The number of spikes per MCC over time should be quantified. Do spike number or intensity correlate with the timing of insertion?

In the revised version of the manuscript, we provide quantification of the number of calcium transients in MCCs during apical emergence (Figure 1E). We observe that the majority of MCCs present more than one calcium transient during apical emergence (see below).

We do not observe any correlation between calcium transients' frequency and/or intensity with the timing of MCC epithelial insertion. Apical emergence is a cell autonomous process, and each MCC will be exposed to a different mechanical environment at the time of epithelial insertion. Therefore, correlating the calcium transient intensity with the timing of insertion while the process varies between individual cells is not possible.

Do calcium transients also concern other radially intercalating cells, in particular ionocytes that insert soon after MCCs?

We thank the reviewer for this insightful comment, which prompted us to examine the appearance of calcium transients in other apically emerging cell types. Live imaging of stage 22 embryos, during which most of the MCCs have inserted into the skin epithelium, revealed that cells adjacent to MCCs, which represent Ionocytes inserting the superficial epithelium, display calcium transients. These data are presented in Movie 11 of the revised manuscript.

It is unclear how the shapes of the apical area in F' are defined based on the views in F.

These data are presented in Figure 1G of the revised manuscript. To help the reader understand how the shape of the apical area in Figure 1G' we have revised Figure 1H to include the outlines of the apical cell surface (see below).

We did not find how the normalization of the surface area was done.

The surface area for time-lapse experiments was normalized by dividing the surface area of each time point by the surface area of the first time point. This description has now been added in the methods section of the revised manuscript.

Fig1G is not convincing with only 3 cells showing significant expansion, considering that apical area expansion is expected to go up with time.

These data are now presented in Figure 1H. For these data, we quantified the surface area increase in a 30-minute time window before a calcium transient and surface area increase in a time window of 30 min after a calcium transient in the same cell. Paired t-test analysis showed that the surface area increase was statistically significantly different when a calcium transient was present. However, as stated above, apical emergence is a cell-autonomous process, and each calcium transient is not expected to be translated to the same degree of surface area expansion. Thus, surface area increase in different cells varies in Figure 1H.

To better examine the correlation between calcium transients' appearance and surface area expansion, we quantified surface area increase in MCCs that displayed or didn't display a calcium transient. This analysis showed that only MCCs displaying a calcium transient display a large increase in their apical surface area. These data are presented in Figure 1I of the revised manuscript (see below).

We do not understand how surface area can be defined and measured in H'', based on the views in H, since cell borders are not visible.

These data are now presented in Figure 1J of the revised manuscript.

In the revised version of the manuscript, we provide close-up images in Figure 1J to better visualise cell borders. (see below).

2. The study is confusing as to whether apical insertion and/or apical expansion depend on calcium signalling. We do not understand the choice of protocol, where drugs are applied at stage 15, several hours before epithelial integration and the functional consequences evaluated at stage 24, while transients are shown to last 20 seconds on recordings done at stage 19. We suggest that calcium transient recording and 2APB treatment should be done at the stage when apical insertion starts, and at the stage when apical expansion is underway.

We apologize that we did not state clearly why we chose this protocol to treat embryos with inhibitors. Our choice of protocol was based on previously published work studying apical emergence or morphogenetic events within the skin epithelium of *Xenopus* embryos occurring at the same developmental period. Specifically, published work

(Colins et al. 2021) focusing on the role of tubulin acetylation during MCC apical emergence treated embryos from stage 13 to stage 26 with nocodazole to examine the effect of nocodazole treatment on apical emergence. Furthermore, Chuyen et al 2021, treated embryos with CK666 and SMIFH2 from stage 12 to stage 25 to examine the effect on MCC apical emergence and MCC spacing, a process completed between stages 22 and 25. Additionally, Kulkarni et al. 2021 treated embryos from stage 14-15 until stage 28 with an inhibitor against mechanosensitive channels to examine its effect on centriole number scaling in MCCs, a process which takes place after initiation of apical emergence and is completed at stage 28. We now cite these papers in the relevant section of the paper.

Calcium transient recordings were done at stages when apical emergence starts (Figure 1B) and during apical surface area expansion (Figure 1G).

In order to address the Reviewer’s comment regarding the timing of 2APB treatment, we treated embryos with 2APB starting at stage 19 when apical emergence begins and at stage 22, mid-apical emergence when apical surface expansion takes place. These experiments allowed us to strengthen our claim that calcium transients are necessary for both the insertion of MCCs into the overlying epithelium and the subsequent expansion of their surface area. Specifically, treatment of embryos with 2APB from stage 19 to stage 24 and from stage 22 to stage 33 resulted in defective MCC apical surface area expansion, with MCCs displaying a smaller surface area. These data are presented in Figure 2F-G and Supplementary Figure 3G-I (see below). See also the relevant response to Reviewer 2.

Incidentally, Fig4B,C seems to report PLC pulses in a cell that has already significantly expanded its surface.

In the initial version of the manuscript, we presented a MCC from a stage 21 embryo during the apical cell surface expansion phase since calcium is necessary for both epithelial insertion and apical cell surface expansion. We understand that this might be confusing. To address the Reviewer's concern, we performed new live imaging experiments of stage 19 embryos. These experiments allowed us to record PLC activity in a MCC inserting into the epithelium and subsequently expanding its surface area. Specifically, these data show pulsed activity of PLC during MCC insertion into the skin epithelium and during MCC apical surface area expansion. Therefore, PLC is active both during the insertion of cells in the epithelium and the subsequent apical expansion. These data are presented in Figure 4A-B (see below) and Movie 7 of the revised manuscript.

3. In relation to the proposed function of calcium transients, it appears important to address whether intercalation and surface expansion is permanently blocked or simply delayed by the various drugs against calcium pathway components. This would inform whether calcium signalling is an absolute requirement for epithelial integration, or if it primarily controls the timing of this process.

To address this comment, we treated embryos with 2APB from stage 15 until stage 28, much later than the completion of MCC apical emergence. These data are presented in Suppl. Figure 3A-C (see below).

This experiment shows that in the absence of calcium MCCs are prevented from apically emerging and not simply delayed.

Similarly, the expression of Calmodulin dominant negative in MCCs results in defective apical emergence even when stage 32 embryos were examined (Figure 5F-H).

Additionally, we provide new data in the revised version of the manuscript showing that overexpression of PLC dominant negative results in defective MCC apical emergence when stage 30 embryos are examined. These data are presented in Supplementary Figure 6 (see below).

This question applies to both MCCs and other intercalating cell types, as we can see only goblet cells in Fig2A (2APB treatment).

We would like to thank the Reviewer for this insightful comment, which allowed us to examine the role of calcium transients in other apically emerging cell types (Ionocytes and Small secretory cells).

To assess the impact of 2APB treatment on Ionocytes and Small secretory cells apical emergence, we treated embryos with 2APB at different stages of development. Specifically, for assessing the impact on Ionocytes apical emergence we treated embryos with 2APB from stage 19 to stage 26. For SCCs apical emergence, we treated embryos with 2APB from stage 27 to stage 33. These experiments allowed us to show that intracellular calcium transients are necessary for both ISCs and SCC apical emergence. These data are presented in Figure 8 of the revised manuscript (see below). A relevant section is added in the results section of the revised manuscript.

4. One major problem with the study as it stands is that functional analyses rely only on drugs and dominant-negative constructs. No attempt was made to deplete critical endogenous components of the pathway. A quick survey reveals that Calmodulin (Calm1) is specifically up-regulated in mature *Xenopus* epidermal MCCs (Lee et al., 2023). It would significantly strengthen the study to include Calm1 knockdown to further inform what is the function of calcium signalling in MCCs.

Calmodilin is encoded by synonymous genes in all vertebrates. Using morpholino oligonucleotides (MOs) to study calmodulin in *Xenopus* is challenging because calmodulin is encoded by four synonymous genes: **calm1s**, **calm1l**, **calm2s**, and **calm2l**, which all produce identical proteins. Therefore, genetic ablation of calmodulin cannot be used to study its function during *Xenopus* development.

Specifically, *Xenopus laevis* Calm2S and Calm2L have identical 5'UTRs and identical sequences after the start codon (see image below), therefore, a single translation-blocking morpholino can be used for these 2 genes.

```

Query 61 TCGATTATACTAACGCGACTATTCCGAAATGGCTGACCAACTGACAGAAGAGCAGATTGC 120
          |||
Sbjct 70 TCGATTAAACTAAGGCAACTATTCCGAAATGGCTGACCAACTGACAGAAGAGCAGATTGC 129
  
```

On the other hand, Calm1L and Calm1S have 3 mismatches at the 5'-UTR and after the start codon. Therefore, 2 different morpholinos will be needed for the knockdown of Calm1L and 1S gene products.

Score	Expect	Identities	Gaps	Strand
1321 bits(715)	0.0	800/840(95%)	10/840(1%)	Plus/Plus

```

Query 144 CGAGACCGCCATGGCTGATCAACTGACAGAAGAACAGATTGCTGAATTTAAGGAGGCTTT 203
          |||
Sbjct 84  CGAGACCACTATGGCCGATCAACTGACAGAAGAACAGATTGCTGAATTTAAGGAGGCTTT 143
  
```

Thus, overall, for knockdown of the synonymous Calmodulin genes in *Xenopus laevis* we would need a morpholino for CALM1S, a morpholino for Calm1L and a morpholino for Calm2S/L. Use of 3 different morpholinos for knockdown of endogenous Calmodulin in *Xenopus* is not an experimentally feasible option.

5. The last section on the link between calcium and actin network formation appears as one of the weakest of the study. The potential implication of RhoA is quickly dismissed with a single case and no quantification.

To address the Reviewer's concern, we quantified the localization of active RhoA at basal bodies by calculating the ratio of the basal bodies signal intensity vs the cortical signal intensity, as described before (Sedzinski et al., 2017). This analysis revealed that RhoA activity during apical emergence is not affected by 2APB treatment. These data are presented in Figure 7B of the revised manuscript (see below).

The single case phalloidin phenotype in Fig7C-E (again no quantification) is extremely mild compared to those reported in Fig. 6F-N.

The data presented in Figure 7C-E of initial submission are now presented in Supplementary Figure 9A,B,D. The impact on apical actin enrichment is evident as indicated by the intensity profile provided in Supplementary Figure 9B. To further support our conclusion we quantified apical actin enrichment in control and 2apb-treated MCCs from this experiment by calculating the ratio of the mean apical actin intensity of a MCC divided by the mean apical actin intensity of 5 neighbouring goblet cells. These data are presented in Supplementary Figure 9 A-C(see below).

Fig7G-I does not help to establish a mechanism, but simply confirms that combining two treatments weakening actin filament formation have more effect than each individual one.

We respectfully disagree with this reviewer's comment. We believe that the data presented in Figure 7F–H go beyond demonstrating additive effects. Rather, they provide critical mechanistic insight into the role of calcium signaling in maintaining the stability of the apical actin network in MCCs.

Specifically, treatment with 2APB alone does not cause a significant disruption of the apical actin architecture. Similarly, 10-minute exposure to Latrunculin A alone, also does not significantly affect the apical actin network. This is consistent with the fact that Lat inhibits the formation of new filaments but has a limited impact on already polymerized filaments.

However, the combination of 2APB and Lat for 10 minutes leads to a dramatic disruption of the apical actin network, as shown in Figure 7F–H. This synergistic effect reveals the necessity of calcium signalling in maintaining the stability of the apical actin network. In the absence of ongoing polymerization (due to Lat), a non-stable actin network (2APB treated MCCs) will be disrupted in a short time period. Therefore, the data in Fig. 7F–H provide key mechanistic evidence that calcium is important for the maintenance and stability of the apical actin cytoskeleton in MCCs. We hope this clarification addresses the reviewer's concern.

To further confirm the data presented in Fig7F-H and address the Reviewers' concern we analyzed actin network dynamics in MCCs from stage 22 control and 2APB treated

embryos expressing mKate-actin using FRAP. FRAP analysis revealed that the fluorescent recovery of mKate-actin is significantly faster in 2apb-treated MCCs (see below), confirming that the apical actin network is more dynamic and less stable in the absence of calcium transients. These data are presented in Suppl. Figure 9F-G and Movie 12 of the revised manuscript.

Here again, the paper would dramatically benefit from the identification of the endogenous link between calcium and actin (Cobl is a good candidate that is up-regulated in mature *Xenopus* MCCs). This would possibly offer the possibility to overcome defects induced by calcium drugs by overexpression of the most downstream effector of the pathway.

We appreciate the reviewer's suggestion and fully agree that identifying the endogenous link between calcium signalling and the actin network during MCC apical emergence would significantly enhance our understanding of the process. However, we believe that uncovering the downstream effectors of the PLC/Ca²⁺/Calmodulin pathway during apical emergence is a project on its own that falls beyond the scope of this revision.

Cobl is indeed an interesting candidate. However, according to expression data from Xenbase, Cobl protein and mRNA start to be expressed stage 22, which is after apical emergence initiation (see below). This expression pattern aligns with the role of Cobl in the function of mature ependymal ciliated cells, as previously described (Mahuzier et al., 2018). This is also in agreement with our preliminary data. Specifically, using a

splicing-blocking morpholino against Cobl, we observed phenotypes in late stages of ciliogenesis, while MCC apical emergence appeared unaffected.

We intend to identify downstream effectors of the PLC/Ca²⁺/Calm signaling axis in future work. This will require the identification of calmodulin-interacting proteins and Ca²⁺/Calmodulin activated proteins present during apical emergence, followed by functional analyses to determine their roles. This work cannot be adequately fulfilled through loss-of-function experiments on individual, already-known CaM interactors alone.

We hope the reviewer will appreciate that while this is an interesting and important direction, addressing it rigorously will require a dedicated project beyond the scope of the current manuscript.

Minor issues:

- Movies must be annotated to help the reader understand the contents. Time should be displayed. We could not identify captions for the movies.

We have annotated the movies as suggested.

- Movie 1 is bugged with a reiteration at about 17s

We don't understand why this happened, but we believe that now it is fixed.

- Many typos are present throughout all sections of the manuscript and in figure legends.

We revised the manuscript and corrected typing mistakes.

- In many places, more care should be given to the wording; e.g. legend 2H mentions the various steps of epithelial insertion, while Fig2H describes epithelial insertion as one of the discrete steps.

We now refer to the whole process as apical emergence, following Reviewer's 2 suggestion, and this helped distinguish the process from the distinct steps.

- The mat/Met section is very minimalist, lacking for instance all commercial references for the various drugs used in the study.

We expanded the Materials and Methods section in the revised manuscript, and we have included all the commercial references for the various drugs used.

- In quite a few places, the authors make logical leaps that are problematic. Fig3D does not establish that PACR-positive cells are MCCs.

In the revised manuscript (Figure3D, see below), we present data showing that PACR-positive cells are MCCs (acetylated tubulin positive).

Fig5F does not establish that CALM-positive cells are MCCs, particularly those on the left column (bottom arrow must be repositioned).

We provided new images in the revised manuscript (Figure 5F, see below) showing that CALM-positive cells are MCCs (acetylated tubulin positive).

In Fig7A the authors use rGBD staining as a proxy for basal bodies, which is not acceptable.

We apologize for this confusion. To measure the number of basal bodies, we used centrin-RFP signal. Representative images are now shown in Figure 7C(see below) of the revised manuscript.

REVIEWER COMMENTS

Reviewer #1 (Remarks to the Author):

The authors have added a number of experiments that have improved the manuscript.

I remain concerned that much of their data is entirely dependent on the 2-APB reagent. Their results do suggest that Calcium plays a critical role; however, whether it is calcium transients is entirely dependent on the 2-APB being specific to calcium transients and not basal calcium levels. They do add data suggesting that basal calcium levels appear unaffected.

Therefore I am inclined to accept their results and see if others can build on their results and augment the evidence.

We thank the reviewer for appreciating the improvement of our manuscript. We understand the reviewer's concern regarding the 2-APB. To address this, we have provided data demonstrating that 2APB treatment blocks calcium transients but does not affect basal calcium levels, leading to defective apical emergence. This is a strong indication that calcium transients are necessary for MCC apical emergence. In support, blockage of calcium transients upon PLC inhibition also results in defective MCC apical emergence. Together these findings strengthen the argument that calcium transients are necessary for apical emergence.

We are pleased that the reviewer is inclined to accept our results, and we look forward to seeing others built on our work in the future.

Reviewer #2 (Remarks to the Author):

This is a much improved paper and I am generally supportive. However, there are still some unresolved issues.

We thank the reviewer for being supportive of our work. According to Reviewer's comments we have revised Figures 4,5,6 and Supplementary Figure 9. In addition, we provide new quantifications in Figure 4I and 6K to address the issues raised by the Reviewer.

1. "Intensity profiles are used widely in the literature...." They are used widely when done properly. The paper has now quantified some of the line scan intensity profiles which is an improvement. However, the "data" in figure 5A-B and 6K are still meaningless.

To address the Reviewer's comment, we have removed the intensity profiles as suggested (Figure 5B, Supplementary Figure 9B) or we have replaced the intensity profile with more meaningful quantification (Figure 6K). See below for detailed responses.

In 5A-B they draw a line and show the profiles of the fluorescent intensity as if this is important, but one could draw a different line in the same cell and get a totally different profile that does NOT show actin/calm between two centrioles. This is minor point but I think 5B should be removed.

To address the Reviewer's comment, we have removed the intensity profile from Figure 5. We don't believe that this is affecting the presented data and the accompanied conclusions.

6K is the same situation, an n of 1 is not an experiment. You know how to quantify fluorescent intensity as you have done in many of the other figures. If you want to include the tubPACR data then do it right or remove it.

We have removed the intensity profile from Figure 6K and we replaced it with quantification of apical actin enrichment in control and atub:PACR expressing MCCs (see below), consistent with panels 6H,M and O.

S9C meaningful, S9B not meaningful.

We have removed the intensity profile from S9B since our conclusions are supported by the graph presented in Figure S9C (Figure S9B in the revised manuscript).

2. In my initial assessment I commented on how sloppy Figure 6D was. I thought it was obvious but clearly I did not explain my concern. Please see the images attached which the authors claim are the same image but with Utr (left) and Geocode (right). Clearly they are not aligned properly. See the opacity overlay to see how out of sync they are.

See attached figure.

We apologize for this unintentional mistake. For transparency, the relevant time lapse recording was presented in Movie 10. We replaced Figure6D with the correct images.

3. My concern regarding figure 4H (now 4G) remains. I originally tried to find the definition of apical insertion because of this image. Now that you have defined apical insertion as $10\mu\text{m}^2$, I am even more concerned. For the image on the right, if your scale bar is accurate, the MCCs is going to be over $10\mu\text{m}^2$ in my estimation. Its $25\mu\text{m}$ long and at least 1 wide. I am not saying that this cell has inserted properly by any means but this representative image calls into question the methodology.

We thank the Reviewer for raising this point that we missed in the previous revision

round. We apologize for not annotating correctly the cells in Figure 4G and failed to explain why we used these exemplary images. These images present MCCs expressing PLCD-PH that either completely failed to insert the superficial epithelium (left example) or failed to expand their surface area(right example). We state this clearly now in Figure 4G (see below).

In the graph presented in Figure 4H, all the cells that had a surface area greater than 10 μm^2 were considered inserted. To further address the Reviewer's comment, we in Figure 4I we present quantification of the surface area of control MCCs and PLCD-expressing MCCs (see below), that have inserted into the epithelium(inserted MCCs from Figure 4H), to highlight the defect of apical surface area expansion in these cells. We believe that quantification of the successful epithelial insertion and of the apical surface area of MCCs expressing PLCD-PH gives the overall picture of the phenotype.

Reviewer #3 (Remarks to the Author):

I would like to commend the authors for the extensive revision work with several new decisive experiments that help strengthen their conclusions. I find particularly important the new demonstration that 2APB also inhibits apical emergence of other

basally localized cells (ISCs and SSCs), which indeed suggests that calcium transients may be more generally involved in epithelial integration in development and homeostasis.

All of my issues have been satisfactorily addressed. I do understand the challenge represented by my request to identify endogenous players by genetic knockdown, and do not consider this a blocking point, considering the numerous improvements made to answer the questions raised by all three reviewers. The paper has been greatly improved and now represents an original and convincing piece of work that should be of interest to many cell and developmental biologists.

We thank the reviewer for appreciating our efforts to revise the manuscript. The Reviewer's suggestions were constructive and helped us to significantly improve the manuscript.